



# Modelling of Multi-Frequency Microwave Backscatter and Emission of Land Surface by a Community Land Active Passive Microwave Radiative Transfer Modelling Platform (CLAP)

Hong Zhao[1], Yijian Zeng[1], Jan G. Hofste[1], Ting Duan[1], Jun Wen[2], Zhongbo Su[1]

1. Faculty of Geo-information Science and Earth Observation (ITC), University of Twente, 7514
   AE, Enschede, The Netherlands

2. College of Atmospheric Sciences, Plateau Atmosphere and Environment Key Laboratory of
   Sichuan Province, Chengdu University of Information Technology, Chengdu, China

*Correspondence to*: Hong Zhao (h.zhao@utwente.nl)

## Abstract

Emission and backscattering signals of land surfaces at different frequencies have distinctive responses to soil and vegetation physical states. The use of multi-frequency combined active and passive microwave signals provides complementary information to better understand and interpret the observed signals in relation to surface states and the underlying physical processes. Such a capability also improves our ability to retrieve surface parameters and states such as soil moisture, freeze-thaw dynamics and vegetation biomass and vegetation water content (VWC) for ecosystem monitoring. We present here a prototype Community Land Active Passive Microwave Radiative Transfer Modelling platform (CLAP) for simulating both backscatter ($\sigma^0$) and emission ($T_B$) signals of land surfaces, in which the CLAP is backboned by an air-to-soil transition model (ATS) (accounting for surface dielectric roughness) integrated with the Advanced Integral Equation Model (AIEM) for modelling soil surface scattering, and the Tor Vergata model for modelling vegetation scattering and the interaction between vegetation and soil parts. The CLAP was used to simulate both ground-based and space-borne multi-frequency microwave measurements collected at the Maqu observatory on the eastern Tibetan plateau. The ground-based systems include a scatterometer system (1-10 GHz) and an L-band microwave radiometer. The space-borne measurements are obtained from the X-band and C-band Advanced Microwave Scanning Radiometer 2 (AMSR2) radiation observations. The impacts of different vegetation properties (i.e.,


structure, water and temperature dynamics) and soil conditions (i.e., different moisture and temperature profiles) on the microwave signals were investigated by CLAP simulation for understanding factors that can account for diurnal variations of the observed signals. The results show that the dynamic VWC partially accounts for the diurnal variation of the observed signal at the low frequencies (i.e., S- and L-

bands), while the diurnal variation of the observed signals at high frequencies (i.e., X- and C-bands) is more due to vegetation temperature changing, which implies the necessity to first disentangle the impact of vegetation temperature for the use of high frequency microwave signals. The model derived vegetation optical depth $\tau$ differs in terms of frequencies and different model parameterizations, while its diurnal variation depends on the diurnal variation of VWC regardless of frequency. After normalizing $\tau$ at multi-

frequency by wavenumber, difference is still observed among different frequencies. This indicates that $\tau$ is indeed frequency-dependent, and $\tau$ for each frequency is suggested to be applied in the retrieval of soil and vegetation parameters. Moreover, $\tau$ at different frequencies (e.g., X-band and L-band) cannot be simply combined for constructing accurate long time series microwave-based vegetation product. To this purpose, it is suggested to investigate the role of the leaf water potential in regulating plant water use and

its impact on the normalized $\tau$ at multi-frequency. Overall, the CLAP is expected to improve our capability for understanding and applying current and future multi-frequency space-borne microwave systems (e.g. those from ROSE-L and CIMR) for vegetation monitoring.




## 1. Introduction

Passive microwave remote sensing instrument (radiometers) for land monitoring measures the amount of microwave radiation that is naturally emitted by the surface, whereas active instruments (scatterometers and radars) transmit a specific microwave signal and measure the backscatter signal of land surfaces (Ulaby et al., 2014a). When only a bare soil surface is considered, the dominant factors that determine the passive emission and active backscatter signals are soil surface roughness and the dynamic soil moisture

and soil temperature profiles (Choudhury et al., 1979; Raju et al., 1995; Ulaby et al., 1981). For vegetated land, the vegetation imposes the effect mainly by contributing its own scattering and emission and attenuating underlying soil scattering and emission (Jackson and Schmugge, 1991; Ferrazzoli and Guerriero, 1996; Ulaby and Wilson, 1985). The presence of the vegetation complicates both components by the interactions of the signals in the vegetation-soil system but also provides the needed signature for

monitoring vegetation.

Utilizing the characteristics of microwaves interacting with vegetation and soil mediums and the existing satellite observations (e.g., Sentinel-1 backscatter, the Advanced Scatterometer (ASCAT) backscatter, the Advanced Microwave Scanning Radiometer for EOS (AMSR-E) and its successor AMSR2, Soil Moisture Ocean Salinity (SMOS) and Soil Moisture Active and Passive (SMAP) radiation), active and passive

microwave remote sensing of land surfaces on a global scale has mainly focused on soil moisture retrieval in the recent past (Su et al., 2020; Dorigo et al., 2017; Fernandez-Moran et al., 2017; Li et al., 2022; Brocca et al., 2017; Bauer-Marschallinger et al., 2018; Gao et al., 2022; Wigneron et al., 2021), mainly due to the crucial impact of soil moisture on the global water, energy and carbon cycles (Humphrey et al., 2021; Zhou et al., 2021; Chatterjee et al., 2022). On the other hand, microwave remote sensing is also

being largely investigated for monitoring vegetation dynamics in natural ecosystems and agricultural applications (Jones et al., 2011; Frappart et al., 2020; Konings et al., 2019; Konings et al., 2021; Steele-Dunne et al., 2017; Prigent et al., 2022). A noticeable application is to utilize the derived vegetation optical depth (VOD or $\tau$, the attenuation effect of vegetation on microwave signal) to estimate vegetation water content for assessing plant status (Konings et al., 2021; Forkel et al., 2022; Wigneron et al., 2020;

Fan et al., 2019; Brandt et al., 2018; Wu et al., 2021).

Despite these advances, the current operational soil moisture and VOD retrieval algorithms mainly rely on zeroth-order radiative transfer theory with numerous empirical assumptions. For instance, in the $\tau$-$\omega$ passive microwave model for the forward simulation and retrievals (Wigneron et al., 2007; De Rosnay et al., 2020; Chan et al., 2015; Li et al., 2022), the effective scattering coefficients $\omega$ is set constant for each

land type and independent of polarizations, vegetation structures and soil moisture, although the dependence of $\omega$ on the foregoing factors and vegetation biome types has been reported (Kurum, 2013;



Konings et al., 2017a; Van De Griend et al., 1996). The parameterized VOD at nadir (τNAD) is generally linked with vegetation indices (e.g., Leaf Area Index (LAI), Normalized Difference Vegetation Index (NDVI) and vegetation water content (VWC)) through empirical equations for low vegetation covers.

Similar site-specific best-fit approaches were also applied for determining surface roughness in the passive microwave case (Wigneron et al., 2017; Chaubell et al., 2020), and for parameterizing vegetation and soil variables in the water cloud model (Attema and Ulaby, 1978) in the active microwave case (Steele-Dunne et al., 2017; Bai et al., 2021).

Other than that the sensor configuration (i.e., frequency, polarization and incidence angle) imposes an

impact on the observed signals. The impacts of the vegetation and surface roughness and their accurate representations in microwave scattering and emission remain unresolved. Consequently, the soil moisture and vegetation parameters estimated from various missions with different model structures and parameters differ in their characteristics and are therefore not consistent (Wang et al., 2019; Zeng et al., 2015). All these results point to a fundamental lack of knowledge in understanding the precise scattering

and emission mechanism of vegetated lands, and the need for in-depth investigations through the forward modelling of microwave backscatter and emission signals at multi-frequency, which can provide complementary and consistent information, in comparison to the ground truth observations.

Great efforts have been made to develop the physically-based scattering-emission model based on Maxwell's equations with the complementary relationship between the emission and scattering (Peake,

1959). The examples include the integral equation model (IEM) (Fung, 1994) and its advanced version (AIEM) (Chen, 2021; Chen et al., 2003) for a rough bare soil surface, and the discrete scattering model (notably Tor Vergata model, hereafter TVG) (Ferrazzoli and Guerriero, 1996; Bracaglia M, 1995) for a vegetated surface. It is known that AIEM assumes isotropic roughness properties for a homogenous dielectric half-space and does not account for the dielectric effects due to heterogeneities in the soil

medium (e.g., composition, moisture content, and bulk density) and the resultant mismatch of impedance between air and soil interface. To account for this, with the aid of comprehensive field observations involving ground-based L-band radiometry observations at an alpine meadow of Maqu site on the Eastern Tibetan Plateau (Su et al., 2020; Su et al., 2011), a physically-based surface dielectric roughness model named the air-to-soil transition (ATS) model (Zhao et al., 2021) has been developed and integrated with

the AIEM that is further coupled with TVG model for modelling L-band scattering and emission of the overall vegetation-soil medium. As such, the coupled ATS-AIEM-TVG model forms the prototype of a Community Land Active Passive Microwave Radiative Transfer Modelling Platform (CLAP) that can be used for integrated modelling, interpretation and application of multi-frequency emission and backscattering signals of land surface.



Progress has been made to understand the scattering and emission processes in vegetated lands at a multi-
annual scale (including freeze-thaw processes) by utilizing the coupled IEM with TVG models and the
Maqu L-band radiometry data (Zheng et al., 2017; Zhao et al., 2021) and Maqu ground-based broad-band
scatterometry data during the winter period (Zheng et al., 2021). A successful retrieval of soil moisture
using Aquarius active and passive microwave data and the foregoing coupled model was achieved (Wang

et al., 2019), which is proven consistent with satellite-derived precipitation and evaporation products on
the Tibetan plateau that is rarely achievable with current operational soil moisture products. Furthermore,
the nature of effective soil temperature and its sampling depth was explained by the in-depth analysis of
the Maqu L-band radiometry data and extended to applications of SMOS and SMAP observation
configurations (Lv et al., 2014; Lv et al., 2016; Lv et al., 2018; Lv et al., 2019).

Despite the above advancement, it is noted that the treatments of vegetation in these studies are based on
the calibration results from Dente et al. (2014), which use both active (ASCAT) and passive (AMSR-E)
microwave signatures of the Maqu area. As a consequence, how the vegetation plays the role in
mechanistic scattering and emission processes in these (*in situ*) microwave observations is not explored
yet. On the other hand, most studies deploy soil moisture (measured or modeled) at the first layer (e.g.,

2.5 cm or 5 cm) for soil emission and backscattering simulations. This is mainly driven by two reasons: 1)
from the theoretical point of view, the penetration depth of soil moisture is about 1/10 of the wavelength
of observation (Wilheit, 1978; Wang, 1987), and 2) from the practical point of view, these depths are the
practical ones where soil moisture sensors (e.g., 5TM, Campbell Scientific Crop) can be installed and
measure soil dielectric constant in the field condition. While it is shown that soil moisture at the real top

layer (e.g., 0 cm) imposes great impacts on variations of sampling depth, especially for high-frequency
signals (Raju et al., 1995; Wang, 1987). It is very hard, if not impossible, to measure it in the field.
Fortunately, the soil process model (e.g., HYDRUS (Hansson et al., 2004), Simultaneous Transfer of
Energy, Mass and Momentum in Unsaturated Soil (STEMMUS) (Zeng et al., 2011a, b; Yu et al., 2018),
with detailed physical representations of soil water and heat transfer and coupling processes can provide

physically consistent soil moisture and temperature profiles with fine discretization in the upper soil layer.
Using the outputs of these mechanistic models as inputs for the radiative transfer model can help
investigate the effect of surface and profile soil states on emission and backscattering signals. It is well
established for crop and tree species that diurnal variations of radar backscatter are due to variations in
plant water dynamics (Brisco et al., 1990; Friesen et al., 2012; Steele-Dunne et al., 2012; Ulaby and

Batlivala, 1976; Vermunt et al., 2021; Konings et al., 2017b; Schwank et al., 2021). The diurnal variation
of L-band emission of grassland during the soil freeze-thaw period is observed being related to soil water
and temperature dynamics (Zheng et al., 2017; Su et al., 2020). However, the variation in dielectric
properties due to the temeprature effect, and how the signal diurnal variation affected by these factors



differ in terms of frequencies and surface condition changes are not fully investigated, due to the lack of
assembled *in situ* long term microwave measurements at multiple frequencies.

Supported by the *in situ* active and passive observations and extracted satellite microwave observations at
the Maqu site, as well as the *in situ* measured and process modeled soil states data, the objective of this
paper is to: 1) use the CLAP for forward simultaneous simulations of both backscatter and emission at
multi-frequency and assess the model performance; 2) investigate how the vegetation structure and
vegetation water and temperature dynamics affect the signal simulation and the derived $\tau$ and $\omega$; 3) probe
the penetration depth of the microwave signals; 4) investigate what factors drive diurnal variations of *in
situ* microwave observations. The improvement of the forward signal modelling is expected to benefit
retrieval methods in the view of monitoring land surfaces from current and future spaceborne radiometer
and SAR observations, such as the Copernicus Imaging Microwave Radiometer (CIMR) (Kilic et al.,
2018), the Copernicus L-band radar observing system for Europe (ROSE-L) (Pierdicca et al., 2019) and
the Water Cycle Observation Mission (WCOM) (Shi et al., 2014).

## 2. Data and methods

### 2.1 Measurements on the Maqu site

The Tibetan Plateau observatory for soil moisture and soil temperature (Tibet-Obs) was built and
maintained since 2006 onwards (Su et al., 2011; Zeng et al., 2016; Zhuang et al., 2020; Zhang et al.,
2021) to provide comprehensive observations for validating reanalysis SM datasets and SM retrievals
from satellite microwave remote sensing (Wang et al., 2018; Dente et al., 2012; Zheng et al., 2018b;
Zheng et al., 2018a; Su et al., 2013; Zeng et al., 2015). Since 2016, an *in situ* Dicke-type radiometer
ELBARA-III at the L-band (1.4 GHz) has been mounted at the Maqu site (33.91°N, 102.16°E) of the
Tibet-Obs, providing long-term brightness temperature $T_B^p$ ($p$ = H or V polarization) observations (with
half-hourly interval) of the land surface (Su et al., 2020). Next to the radiometer tower, the SMST_LC pit
(Lv et al., 2018) installed with the 5TM ECH2O probes (METER Group, Inc. USA) at depths ranging
from 2.5 cm to 1 m, provides profile soil moisture and temperature at 15-minute interval (Su et al., 2020).
The soil samples collected from this pit were also utilized to provide soil texture information (Zhao et al.,
180    2018).

In August 2017, the ground-based scatterometer was installed also on the tower and continued to operate
until December 2018, providing broad-band ranging from L-band (1.5–1.75 GHz), S-band (2.5–3.0 GHz),
C-band (4.5–5.0 GHz), to X-band (9.0–10.0 GHz) backscatter observations (with hourly interval) at the
four linear polarizations (i.e., HH, VV, VH and HV) (Hofste et al., 2021). Considering the effect of the
scatterometer antenna pattern, especially a large pattern angle in the low frequency, Hofste et al. (2021)



derived the effective incidence angle characterized from the antenna to the calculated footprint center, where the surface projected antenna beam intensity is equal to half its maximum value. The incidence angle range (i.e., minimum and maximum angles in Table 1) that determine the footprint area was also obtained for each frequency (Hofste et al., 2021). Although the backscatter at both cross-polarization (i.e.,

VH and HV) were measured, they exhibit the difference in terms of frequency and land surface condition change (Hofste et al., 2021). This paper only focuses on the observation at VH polarization for simplicity. Table 1 summarizes the sensor configuration of the ground observed active and passive microwave signals used in this study. It is to note that there is no *in situ* measured X- and C-bands emission data, the extracted AMSR2 emission data at the descending mode (Table 1) is thus used as a surrogate for

comparison to assess the model performance.

Data of vegetation parameters such as fresh and dry above-ground biomass, vegetation water content (VWC), leaf area index (LAI), and vegetation height are available on two summer days (i.e., 12/06/2018 and 17/08/2018) during the field campaign (please refer to Table A2 in Hofste et al. (2021)). Time series LAI data extracted from MCD15A2H-MODIS/Terra+Aqua Leaf Area Index (500m resolution)

(https://lpdaac.usgs.gov/products/mcd15a2hv006/) and then processed with the harmonic analysis of the time series (HANTS) algorithm (Verhoef, 1996) are also for use (i.e., to calculate the number of scatterers) in this study.

Table 1 Information on the microwave observation on the Maqu site.

| Mode | Sensor name | Band | Centre frequency (GHz) | Effective incidence angle | Incidence angle range | Polarization | Data source |
|---|---|---|---|---|---|---|---|
| A | Ground-based scatterometer | X | 9.5 | 54° | 48°-58° | HH, VV, VH | Hofste et al. (2021) |
| A | Same as above | C | 4.75 | 51° | 39°-60° | Same as above | Same as above |
| A | Same as above | S | 2.75 | 44° | 39°-60° | Same as above | Same as above |
| A | Same as above | L | 1.625 | 40° | 21°-60° | Same as above | Same as above |
| P | Ground-based ELBARA-III radiometer | L | 1.4 | 40° | | H and V | Su et al., 2020 |
| P | AMSR2 | X/C | 6.9/10.7 | 55° | | H and V | NRT AMSR2 Unified L3 Daily 25 km TB & SIC Polar Grids Version 4 |

Where A and P refer to the active and passive modes respectively.



## 2.2 Estimated vegetation water content (VWC)

Vegetation water content is an important parameter determining vegetation dielectric constant. It is known that the diurnal solar radiation cycle not only affects vegetation temperature but also vegetation water content. The mechanism behind this is that during the day, the rate of transpiration exceeds the rate of root water uptake, decreasing the volume of water within vegetation, while water reserves are replenished in the late afternoon to the early morning when the rate of root water uptake exceeds the rate of transpiration (Monteith, 2020; Brisco et al., 1990; Konings et al., 2019). Considering this plant physiology, we used the sinusoidal function below (Eq. (1)) to estimate diurnal VWC during the summer period (at the end of July in this case) with the aid of *in situ* measurements.

$$VWC = VWC\_m + Asin(\omega t + \phi) \qquad (1)$$

Where $VWC\_m$ is the mean vegetation water content (kg/kg), A is the amplitude of the VWC fluctuation (the range from maximum or from minimum to the average temperature). In this case, based on the *in situ* measurement conducted by Hofste et al. (2021), $VWC\_m$ is valued at 0.6, and $A$ is valued at 0.05. $\omega$ is the radial frequency, which is $2\pi$ times the actual frequency. In the case of diurnal variation, the period is 86,400 sec (24 hour), so $\omega= 2\pi/86,400 = 7.27 \times 10^{-5}$/sec. $\phi$ is the phase shift and is valued at $\pi/2$ in radians. The shape of the estimated diurnal VWC is displayed in Fig. S1 in the supplementary materials.

## 2.3 Process-model simulated profile soil moisture and temperature data

As an initial attempt to investigate the effect of the surface (soil moisture) state on the microwave signal simulation, we obtained simulated data of soil moisture and temperature at the depths of 0.1, 1, 2, 5, 10, 15, 20, 30, 40, 50 and 60 cm with the vadose zone process model STEMMUS (Simultaneous Transfer of Energy, Mass and Momentum in Unsaturated Soil) (Yu et al., 2020). The simulated data during the winter period is focused, as we assume that vegetation is dead in this period and the ground surface condition changes (i.e., in moisture and temperature) are the main driving factors for the signal variation. Fig. S2 shows the time series STEMMUS simulated soil moisture and temperature comparable to the *in situ* measurements  (i.e., mean soil moisture of 0.12 vs 0.07 at 2.5  cm, 0.1 vs 0.05 at 5 cm and 0.1 vs 0.09 at 10 cm shown in Fig. S3) during the winter period.

## 2.4 Community Land Active Passive Microwave Radiative Transfer Modelling Platform (CLAP)

As already mentioned, the current prototype of CLAP consists of two main components. One component relies on the TVG model (Ferrazzoli and Guerriero, 1996; Bracaglia M, 1995) for modelling vegetation scattering and the interaction with underlying soil scattering, which includes effects of multiple scattering



both within the vegetation and between vegetation and soil. The other component is the ATS model (Zhao et al., 2021) integrated with the AIEM (Chen, 2021) for modelling soil surface scattering.

In the TVG model vegetation is described as an ensemble of discrete scatterers with assigned geometries, such as a uniform top layer of discs (leaves) and cylinders (stems) with structure symmetrical in azimuth. After defining the shape of discrete scatters, the corresponding electromagnetic approximations are adopted in respect of selected geometry and frequency (Fig. 1) to calculate vegetation bistatic scattering and extinction (absorption plus scattering) cross-sections, in which the vegetation dielectric constant is calculated from either the Matzler (1994) model (gravimetric VWC not less than 0.5 (kg/kg)) or Ulaby (1987) model (dry vegetation). Thereby the scattering and transmission matrices of the scatter layer are computed, in which each element represents the ratio between the specific intensity scattered into an upper (lower) angular interval of scattering off-normal angle $\theta_s$, and the specific intensity incoming from an upper angular interval of incidence off-normal angle $\theta_i$. The scattering dependence on the difference between incidence and scattering azimuth angles $(\varphi_s - \varphi_i)$ is expressed by the Fourier series (Ferrazzoli et al., 1991). A similar Fourier transform is applied for obtaining soil scattering matrix in terms of incoherent bistatic scattering coefficients calculated from the integrated ATS+AIEM model, and coherent specular reflection coefficients computed from the Fresnel equations corrected by a roughness factor. By using the matrix doubling algorithm (Fig. 1), the contributions from all layer scatters are integrated for the whole vegetation, and the contributions from the whole vegetation and from the soil part are also combined.

The scattering coefficients in the backward direction, namely backscattering coefficients from the vegetation part ($\sigma_v^0$), soil part ($\sigma_s^0$) and their interaction ($\sigma_{vs}^0$), respectively, are then obtained through Fourier back-transform with $\varphi_s - \varphi_i = \pi$, and the total backscattering coefficient ($\sigma^0$) is calculated by summing them up (Eq. (2)). The polarization and frequency dependences are suppressed in the equation for brevity.

$$\sigma^0 = \sigma_v^0 + \sigma_{vs}^0 + \sigma_s^0 \qquad (2)$$

The $\tau$ and $\omega$ can be estimated below (Eq. (3)) from the simulated vegetation emissivity $e_v$ and vegetation transmissivity $\gamma_v$ same as in Ferrazzoli et al. (2002).

$$\omega = (1 - \frac{1-\varepsilon_v}{\gamma_v}) \qquad (3a)$$

$$\tau = -\ln(\gamma_v)\cos(\theta_i) \qquad (3b)$$


For the passive part, the emissivity $\varepsilon$ at an incidence angle $\theta_i$ is obtained by applying energy conservation

law by integrating the bistatic scattering coefficients over the half-space above the surface (Fig. 1). Due to

the low vegetation emission, the physical temperature of vegetation is assumed the same as that of soil in

this study. The effective soil temperature $T_{eff}$ is estimated using Lv incoherent model (Lv et al., 2016)

(please refer to section 2.2.3 in Zhao et al. (2021)). Finally, the equation $\varepsilon(\theta_i) \cdot T_{eff}$ is used to calculate

brightness temperature $T_B$. The flowchart illustrating the procedure for the forward simulation of

microwave active and passive signals is presented in Fig. 1.

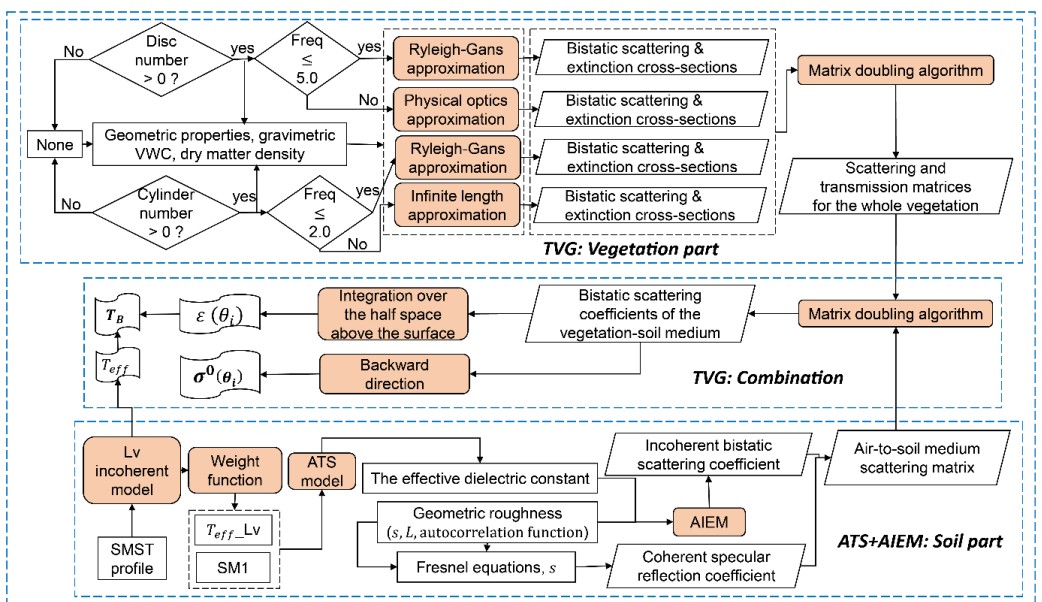

*Figure 1 Flowchart illustrating the procedure for the forward $\sigma^0$ and $T_B$ simulations by CLAP (the*
*coupled ATS-AIEM-TVG model). Freq denotes frequency with a unit of GHz. The square rectangle*
*represents inputs and parameters, and the rounded rectangle in orange refers to models and algorithms.*
*The outermost dash blue box encloses elements in the ATS-AIEM-TVG model. Inside three dash boxes in*
*blue enclose elements in modelling scattering of vegetation, soil parts and their combination respectively.*
*The black dashed box inside the upper blue dash box encloses different vegetation electromagnetic*
*approximation approaches and the corresponding calculated bistatic scattering and extinction cross-*
*sections. The black dashed box inside the lower blue dash box is with inputs used for the ATS model to*
*obtain the effective dielectric constant of the composite air-to-soil medium, in which SM1 refers to soil*
*moisture at the first layer. Detailed descriptions of the ATS model are seen in section 2.2.2 as well as Lv*
*incoherent models in section 2.2.3 in Zhao et al. (2021).*



### 2.5 Configuration of simulation experiments

In the default setting grass leaves at the Maqu site are assumed to be of disc geometries (Dente et al., 2014). Considering *Festuca ovina (Gu et al., 2009; Miller, 2005)* dominating in the study field (please also refer to Fig. A1 in Hofste et al. (2021)), the grass leaves are also parameterized as cylinders in this study for comparison. Table 2 lists the geometric parameters and their values used in the disc- and cylinder- configurations. The values of the cylinder parameters such as radius, and length (inferred from

plant height) and the number of cylinders are *in situ* values measured by Hofste et al. (2021). The Eulerian angles (alpha, beta and gamma) describing leaf orientation (please refer to Fig. 5 in Eom and Fung (1984)) are the investigated values to converge model simulation (see Table 3 in (Hofste et al., 2022)). The values of angular resolutions in the off-nadir angle $\theta$ and azimuth angle $\varphi$ directions for calculating the scatter's bistatic cross-sections are doubled in the cylinder configuration, because for

cylinders high resolutions are necessary, since their scattering patterns are more peaked due to their large dimensions with respective to the wavelength.

Table 2 Parameters in the disc- and cylinder- configurations.

| Disc parameterization | | Reference |
|---|---|---|
| Disc: radius (cm) | 1.4 | |
| Disc: thickness (cm) | 0.02 | |
| Disc: number per unit of area (1/cm$^{-2}$) | LAI/($\pi$ * radius * radius) | |
| Angular resolution in off-nadir angle $\theta$ direction (namely, number of discrete intervals of incidence and scattering off-nadir angles) | 36 | |
| Angular resolution in azimuth angle $\varphi$ direction (namely, number of Fourier components for dependence on $(\varphi_s - \varphi_i)$ of scattering (Ferrazzoli et al., 1991)) | 64 | Dente et al. (2014) |
| Eulerian angles describing leaf orientation (Eom and Fung, 1984) | alpha = 15, 12, 30 beta = 5, 17, 5 gamma = 0, 1, 0, where the 1st number is starting angle, the 2nd is the number of total angles, and the 3rd number is the angular increment. | |
| Cylinder parameterization | | |
| Cylinder: radius (cm) | 0.05 | |
| Cylinder: length (cm) | 30 | |
| Cylinder: number per unit of area (1/cm$^{-2}$) | 2 | |
| Angular resolution in off-nadir angle $\theta$ direction (namely, number of discrete intervals of incidence and scattering off-nadir angles) | 72 | Hofste et al. (2021) Hofste et al. (2022) |
| Angular resolution in azimuth angle $\varphi$ (namely, number of Fourier components for dependence on $(\varphi_s - \varphi_i)$ of scattering (Ferrazzoli et al., 1991)) | 128 | |



| | |
|---|---|
| Eulerian angles describing leaf orientation (Eom and Fung, 1984) | alpha = 0, 45, 8<br>beta = 0, 30, 3 (for summer period), and beta = 90, 1, 90 (for winter period)<br>gamma = 0, 1, 0, where the 1st number is starting angle, the 2nd is the number of total angles, and the 3rd number is the angular increment. |

Considering the different nature of vegetation, soil and surface roughness status during the summer and winter periods, this study carries out model simulations for each period separately. We focused on

simulations with one hour interval during the summer period from 20/07/2018 to 05/08/2018 and during the winter period from 01/01/2018 to 15/01/2018, where the prior information is available (Hofste et al., 2021; Zheng et al., 2021). Accordingly, we also analyzed the performance of microwave multi-frequency signal forward modelling at the diurnal level.

Regarding vegetation scattering modelling, the estimated dynamic VWC in section 2.2 is used during the

summer period, and a constant value of 0.04 (kg/kg) of VWC is applied during the winter period, because the dehydrated dead vegetation is assumed in this period. The dead vegetation is further assumed lying on the ground with a beta angle of 90° (Table 2). The number of cylinders is assumed to be constant over these two periods, considering that the observed scene is in a fenced area and without yak grazing activities (Su et al., 2020). Variation of vegetation temperature is significant in determining liquid water

dielectric constant and associated vegetation dielectric constant (Ulaby et al., 2014b). As there is no *in situ* measured vegetation temperature, the value of air temperature measured at 2 m above the surface (Fig. 2b ) is assigned to the value of vegetation temperature, which is acceptable because of the low air pressure on the Tibetan Plateau.

Regarding soil scattering modelling, the information of soil moisture at the first layer (i.e., 2.5 cm in the

*in situ* case and 1 mm in the STEMMUS output) is fed into the ATS model for calculating the effective dielectric constant of the composite air-to-soil medium (Zhao et al., 2021). For surface roughness parameters—the standard deviation of height *s* and correlation length *L*, the default calibrated *s* of 0.9 cm and *L* of 9 cm (Dente et al., 2014) at the Maqu site are used during the summer period, considering that the calibration is based on satellite observations in the normal (thawed) soil condition. The different

values of *s* of 0.4 cm and *L* of 12 cm are used during the winter period, which were calibrated by Zheng et al. (2021) based on in-situ observations in this period. Finally, the main simulation is conducted at the effective angle of incidence (Table 1) and the results are compared to the observations with the statistics of Bias and root mean square error (RMSE) calculated. The uncertainty in the simulation due to the




footprint effect is quantified through conducting simulations at the minimum and maximum angles of

incidence (Table 1) separately for each frequency.

In the model simulation, during the winter period, the value of the obtained soil backscattering coefficient $\sigma_s^0$ for cross-polarization was often found lower than the equivalent minimum detectable radar cross section of the scatterometer, making it meaningless. This level is derived from the minimum detectable radar cross section of the scatterometer system specified in Table S1 in the supplementary materials.

Simulation values below the minimum detectable level were filtered out from the analysis.

## 3. Results

### 3.1 Simulated backscatter ($\sigma^0$) signals at multi-frequency during the summer period

Figure 2 shows that the cylinder-based simulated $\sigma_{pq}^0$ at the X-band is closer to the observation, especially at VV polarization than those simulated based on the disc parameterization of vegetation (Biases of 0.7

dB vs 3.0 dB, and RMSEs of 1.8 dB vs 3.4 dB in Table 3). The simulated $\sigma_{pq}^0$ at the X-band exhibits diurnal variations mainly due to the consideration of dynamic vegetation temperature (see also Fig. S4, which shows the sensitivity analysis result of signal variation on vegetation temperature and VWC), although the variation does not agree with the observation completely, indicating that the observed signal at the X-band with weak penetration capability may come from the top structure of vegetation, whose

variation is readily driven by wind (see the Fourier transform analysis shown in Fig. S5, where in the frequency domain, both the observed $\sigma^0$ and wind speed variable show fluctuations). By analysing the contribution from the different components, the vegetation contribution is found dominating in both disc- and cylinder-based simulations at all polarizations (Fig. S6), where the difference on average between the cylinder- (disc-) based simulated total signal $\sigma^0$ and the vegetation component $\sigma_v^0$ is 0. (0.6) dB, and the

difference between the cylinder- (disc-) based simulated $\sigma^0$ and the soil component $\sigma_s^0$ is 35.9 (19.5) dB.



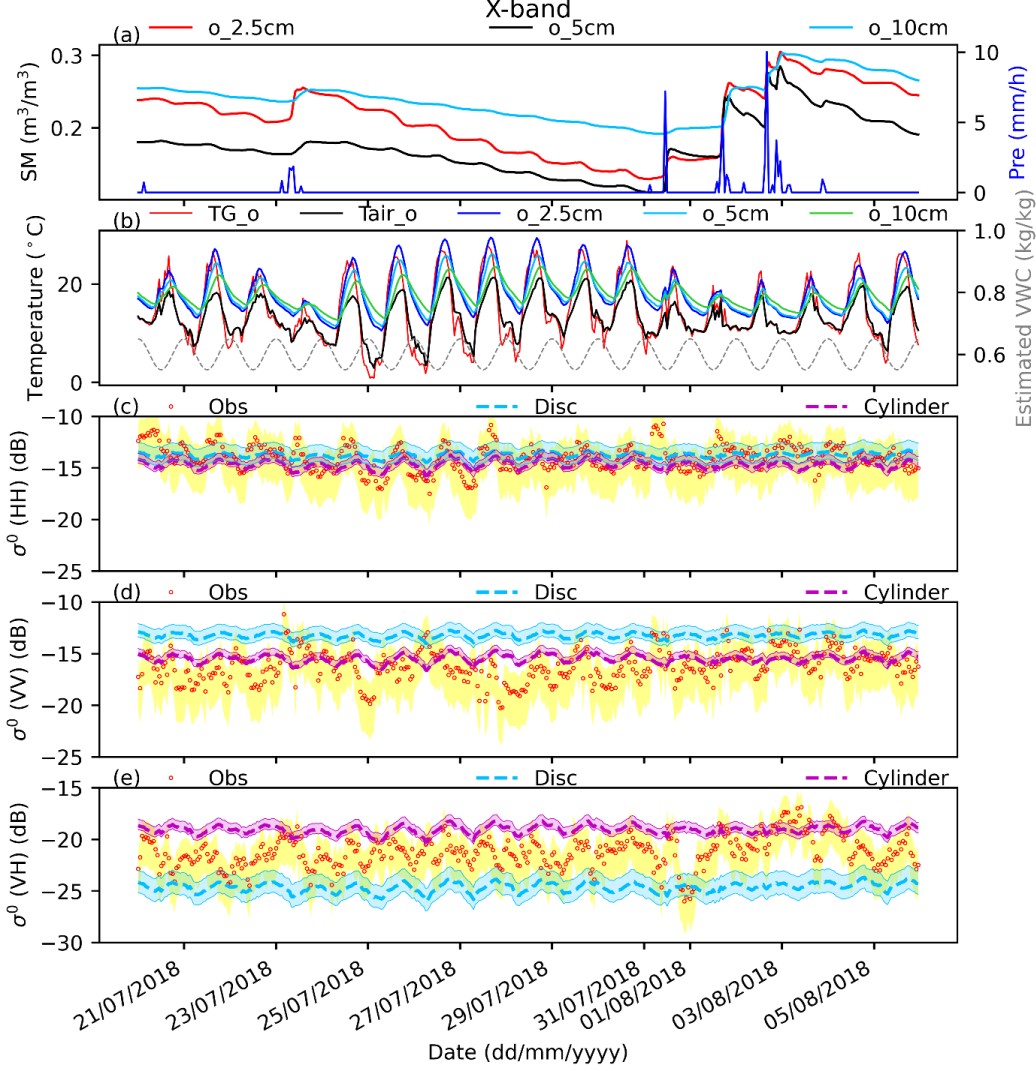

*Figure 2 $\sigma_{pq}^0$ at the X-band simulated by the CLAP with the disc and cylinder parameterizations compared to the ground-based observations during the summer period. Panels a and b display the in situ measured (o_) soil moisture and temperature at different depths and environmental variables during this period, in which TG refers to ground surface temperature, Tair represents air temperature at 2 m, and Pre denotes precipitation. In panels c, d and e, the shaded area overlapping the simulation results refer to the uncertainty due to the effect of the footprint area determined by the different incidence angles (please refer to Table 1), and the yellow shaded area refers to the uncertainty in the observed backscatter data (see Hofste et al. (2021)).*






*Table 3 Statistics of the comparison of $\sigma_{pq}^0$ at multi-frequency simulated by the CLAP using the disc and cylinder parameterizations respectively, to the ground-based observations during the summer period. Bias and RMSE are in the unit of dB.*

| Band | Scheme | HH | | VV | | VH | |
|------|--------|------|------|------|------|------|------|
| | | Bias | RMSE | Bias | RMSE | Bias | RMSE |
| X | Disc | 0.3 | 1.3 | 3.0 | 3.4 | -3.4 | 3.8 |
| | Cylinder | -0.7 | 1.4 | 0.7 | 1.8 | 2.1 | 2.6 |
| C | Disc | 4.1 | 4.3 | 5.8 | 6.0 | -1.1 | 1.7 |
| | Cylinder | -1.2 | 1.7 | 0.5 | 1.5 | -0.3 | 1.1 |
| S | Disc | 6.0 | 6.1 | 6.2 | 6.3 | -4.0 | 4.2 |
| | Cylinder | 2.9 | 3.0 | 1.8 | 2.1 | -2.0 | 2.3 |
| L | Disc | -1.5 | 1.8 | 5.8 | 5.9 | -9.9 | 10.0 |
| | Cylinder | -4.7 | 4.8 | 3.9 | 4.1 | -32.1 | 32.2 |

Figure 3 (C-band) and Figure 4 (S-band) also show that the cylinder parameterization performs better (lower Biases and RMSEs (e.g., RMSEs of 1.5 dB vs 6.0 dB in Table 3) than the disc parameterization in

$\sigma_{pq}^0$ simulations, especially at VV polarization. In the simulation of $\sigma^0$ at the C-band at all polarizations, the vegetation contribution still dominates in both simulations (Fig. S7), where the difference on average between cylinder- (disc-) based simulated $\sigma^0$ and $\sigma_v^0$ is 2.5 (0.6) dB, and the difference between cylinder- (disc-) based simulated $\sigma^0$ and $\sigma_s^0$ is 16.7 (21.8) dB. While in the simulation of co-polarization $\sigma^0$ at the S-band, the soil contribution is found dominant in both simulations (Fig. S8), where the difference on

average between cylinder- (disc-) based simulated $\sigma^0$ and $\sigma_s^0$ is 2.7 (2.3) dB, and the difference between cylinder- (disc-) based simulated $\sigma^0$ and $\sigma_v^0$ is 4.9 (5.5) dB. Comparatively, both $\sigma_v^0$ and $\sigma_{vs}^0$ are dominant in both simulations of $\sigma_{VH}^0$ at the S-band (Fig. S8), and the mean difference is 2.6 dB between simulated $\sigma_{VH}^0$ and $\sigma_v^0$ and 3.9 dB between simulated $\sigma_{VH}^0$ and $\sigma_{vs}^0$). Moreover, consistent diurnal variations are found between the simulated $\sigma_{pq}^0$ at the C- and S-bands and the observations (Figs. 3 and 4), especially at

HH polarization. Based on the results of sensitivity analysis of signal variation on vegetation temperature and VWC shown in Fig. S9, it is observed that the impact of dynamic vegetation temperature is larger than that of dynamic VWC in simulating C-band microwave signal with diurnal variations (also see Figs. 2a and 2b with Fig. 3), which is consistent with the foregoing at the X-band. While it is the other way around, namely the dynamic VWC rather than vegetation temperature contributes more to the diurnal

variations of the observed $\sigma^0$ at the S-band (Fig. S10).

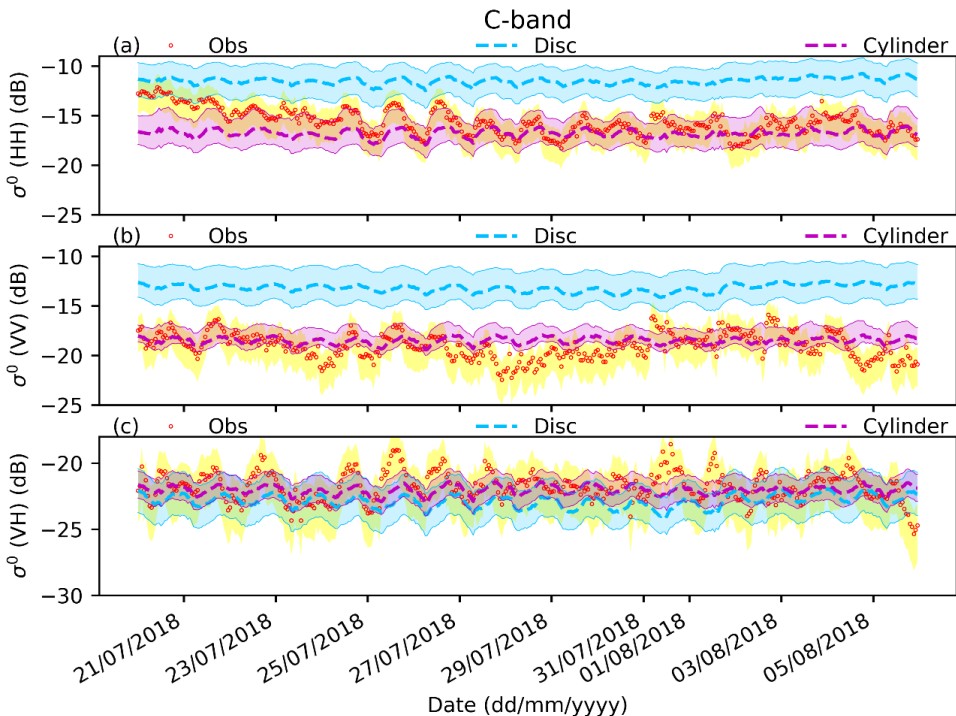

*Figure 3 Similar to Figure 2 but for C-band. The plot of in situ soil moisture and temperature at different depths and environmental variable observations in this period can be found in Figures 2a and 2b.*





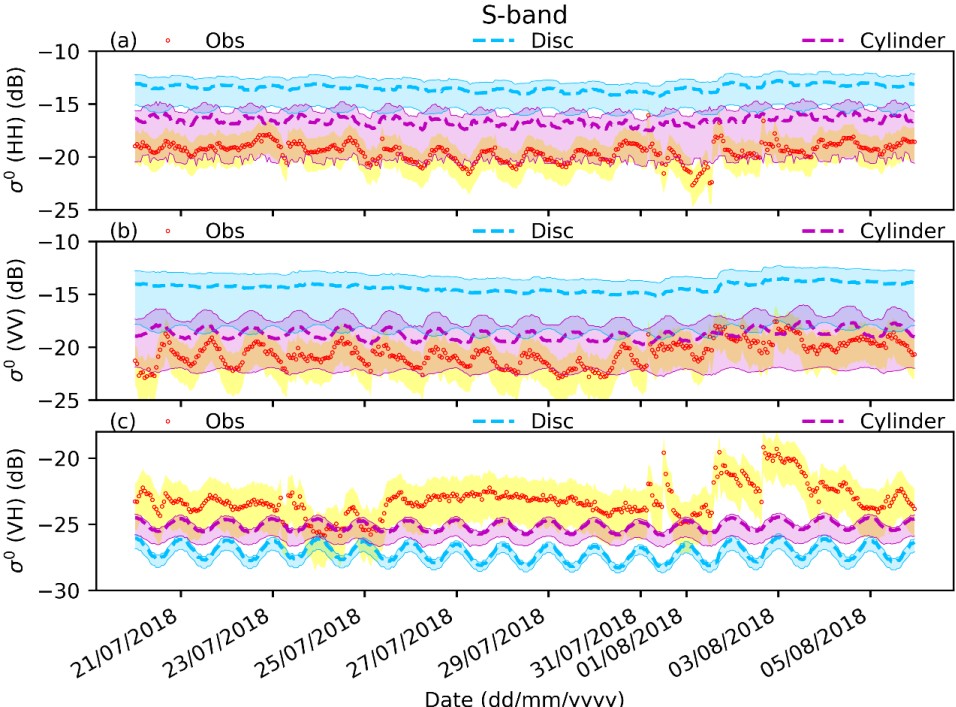


*Figure 4 Same as Figure 2 but for S-band.*

Figure 5 shows that the disc parameterization performs better in simulating $\sigma_{HH}^0$ and $\sigma_{VH}^0$ at the L-band, where the close values of Biases and RMSEs (e.g., RMSEs of 1.8 dB vs 4.8 dB for HH polarization in Table 3) are observed, while a large discrepancy (RMSEs of 10.0 dB vs 32.2 dB in Table 3) is observed between the simulated $\sigma_{VH}^0$ and the observation. Based on the sensitivity analysis of signal variation on vegetation temperature and VWC shown in Figs. S11 and S12, the diurnal variations of the simulated $\sigma_{HH}^0$ and $\sigma_{VH}^0$ are observed mainly due to the dynamic VWC rather than the vegetation temperature reported for X- and C-bands (Figs. S4 and S9). Comparably, better performance of the cylinder parameterization is still found in the signal simulation at VV polarization (Fig. 5, Biases of 3.9 dB vs 5.8 dB and RMSEs of 4.1dB vs 5.9 dB in Table 3), and the use of dynamic VWC results in simulated $\sigma_{VV}^0$ closer to the observation than using constant VWC does (see the sensitivity analysis result shown in Fig. S12). The soil contribution is found dominating in the cylinder-based simulation at all polarizations with a mean difference of 1.6 dB between simulated $\sigma^0$ and $\sigma_s^0$ (Fig. S13). In the disc-based simulation, $\sigma_s^0$ is dominant only at VV polarization, and both $\sigma_v^0$ and $\sigma_{vs}^0$ are dominant at HH and VH polarizations (Fig. S13).

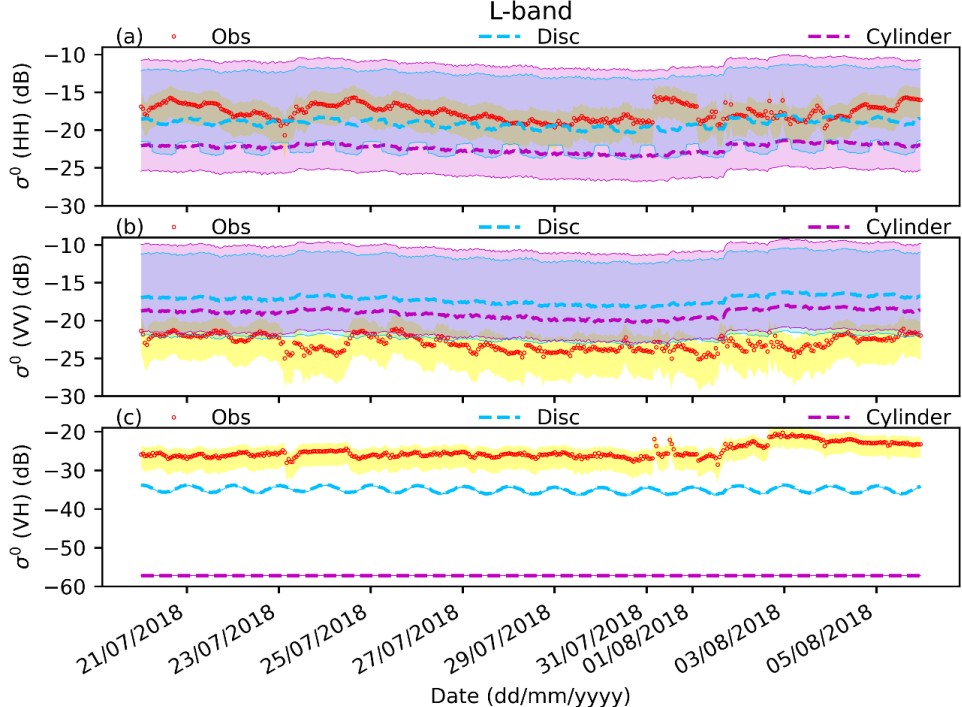

*Figure 5 Same as Figure 2 but for L-band.*

It can be concluded that the CLAP using the cylinder parameterization can mimic the observed $\sigma_{pq}^0$ at multi-frequency well for grassland during the summer period, despite the discrepancy between the simulated $\sigma_{VH}^0$ at the L-band and the observation. Moreover, the CLAP simulation results demonstrate the

characteristic of multi-frequency microwave interacting with soil and grass during the peak growth period (i.e., July and August). Such results indicate that the vegetation scattering dominates in the observed total scattering at high frequencies (i.e., X- and C-bands), while soil scattering itself dominates at low frequencies (i.e., S- and L-bands) at co-polarization, and both vegetation and the interaction between vegetation and soil are dominant for cross-polarization at the low frequencies. Furthermore, by

considering dynamic diurnal vegetation temperature and VWC in the simulation system, the observed diurnal variations of microwave signals can be mimicked well.

## 3.2 Simulated emission ($T_B$) at multi-frequency during the summer period

Figure 6 shows that the disc parameterization performs better in mimicking ELBARA-III observed $T_B^H$ at

the L-band (RMSEs of 12.7 vs 49.7 K in Table 4), while performing similarly as the cylinder

parameterization does in the simulation of $T_B^V$ (RMSEs of 6.5 K and 7.7 K in Table 4). This finding is consistent with those in simulating scatterometer observed co-polarization $\sigma_{pq}^0$ at the L-band (Fig. 5 and Table 3). Figures S14 and S15 show that the simulated $T_B^p$ signals at the X- and C-bands during the summer period can capture the variation of the AMSR2 observed $T_B^p$ signals, and the disc

parameterization results in simulated $T_B^p$ closer to the AMSR2 measurements (lower Biases and RMSEs in Table 4) than the cylinder parameterization does. However, caution should be taken about the different spatial resolution (i.e., *m* vs *km*) between simulated results and the satellite measurements in drawing this conclusion.

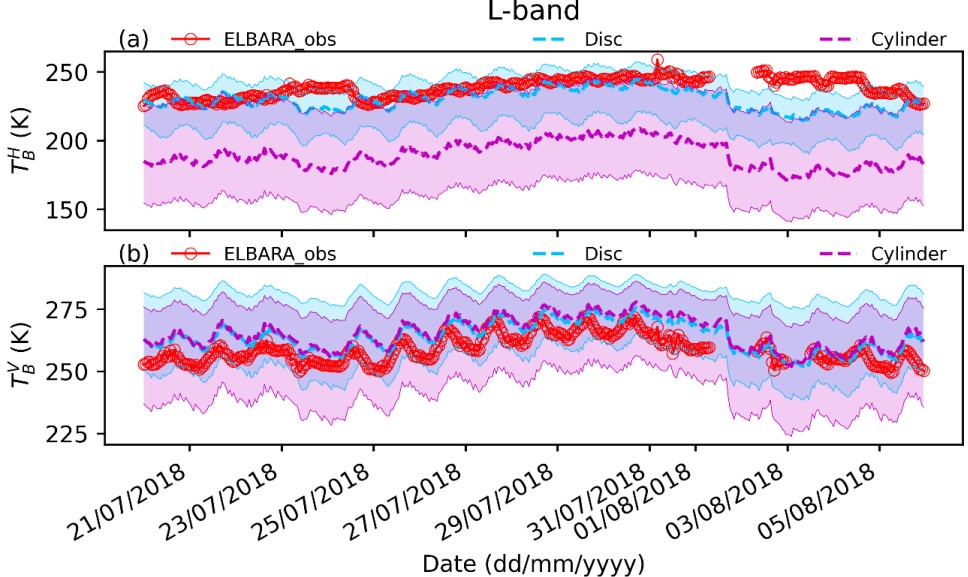

*Figure 6 $T_B^p$ at the L-band simulated by the CLAP with the disc and cylinder parameterizations compared to the ELBARA-III observations during the summer period. Similar to Figure 5, the shaded area overlapping the simulation results refers to the uncertainty due to the effect of the footprint area determined by the different incidence angles (please refer to Table 1). Regarding ELBARA-III observations, the peaks (e.g., from 31/07/2018 to 03/08/2018) due to surface reflected solar beams into*

*the ELBARA-III antenna horn under certain surface conditions such as after rainfall events (Su et al., 2020) are filtered.*

*Table 4 Same as Table 3 but for the comparison of $T_B^p$ (p = H or V polarization). Bias and RMSE are in the unit of K.*

| Band | Scheme | H | V |
|---|---|---|---|

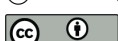



|   |          | Bias  | RMSE | Bias | RMSE |
|---|----------|-------|------|------|------|
| X | Disc     | 0.7   | 5.9  | 15.2 | 2.4  |
|   | Cylinder | 8.3   | 14.0 | 21.7 | 5.8  |
| C | Disc     | -11.7 | 14.7 | 6.0  | 9.2  |
|   | Cylinder | 12.2  | 15.3 | 21.8 | 10.4 |
| L | Disc     | -8.5  | 12.7 | 5.4  | 6.5  |
|   | Cylinder | -48.6 | 49.7 | 6.9  | 7.7  |

### 3.3 Simulated backscatter ($\sigma^0$) at multi-frequency during the winter period

Figure 7 shows that the cylinder-based simulated $\sigma^0_{pq}$ at the X-band is closer to the observation at all polarizations than the disc-based simulated $\sigma^0_{pq}$ does. The cylinder-based simulated $\sigma^0_{VV}$ matches the observation well (RMSE of 1.7 dB in Table 5), while a large discrepancy is observed between simulated $\sigma^0_{HH}$ and the observation (RMSE of 11.5 dB) due to the overestimated $\sigma^0_s$ as shown in Fig. S16. A large discrepancy is also observed for simulated $\sigma^0_{VH}$ against the observation (RMSE of 9.6 dB).

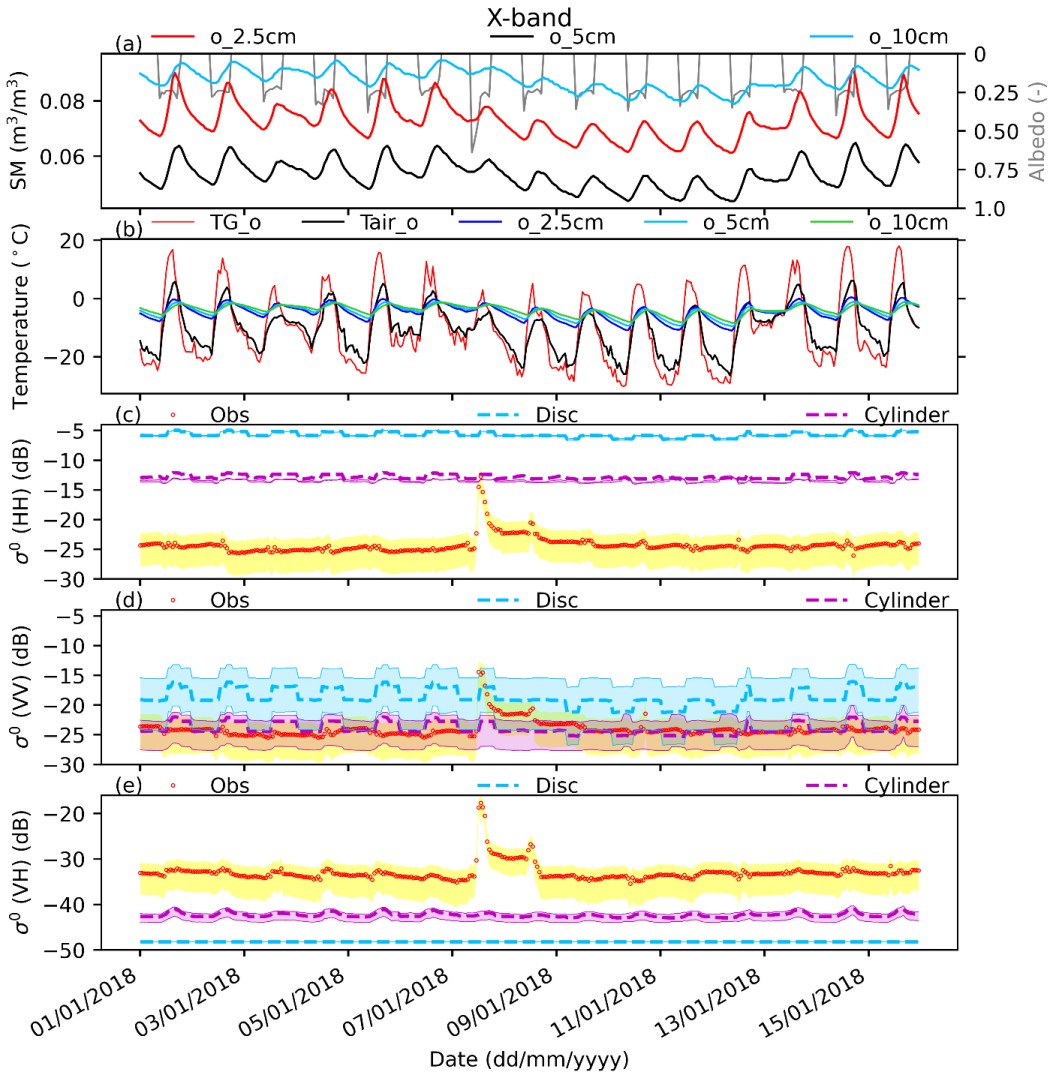

*Figure 7 $\sigma^0_{pq}$ at the X-band simulated by the CLAP with the disc and cylinder parameterizations compared to the ground-based observations during the winter period. Panels a and b display the in situ measured (o_) soil moisture and temperature at different depths and environmental variables during this period, in which TG refers to ground surface temperature and Tair represents air temperature at 2 m. In panels c, d and e, the shaded area overlapping the simulation results refer to the uncertainty due to the effect of the footprint area determined by the different incidence angles (please refer to Table 1, and the yellow shaded area refers to the uncertainty in the observed backscatter data (see Hofste et al. (2021)).*

*Table 5 Same as Table 3 but for the winter period. Bias and RMSE are in the unit of dB.*

| Band | Scheme | HH | | VV | | VH | |
|------|--------|------|------|------|------|------|------|
| | | Bias | RMSE | Bias | RMSE | Bias | RMSE |





| | | | | | | | |
|---|---|---|---|---|---|---|---|
| X | Disc | 18.6 | 18.6 | 5.4 | 5.7 | -15.2 | 15.4 |
| | Cylinder | 11.4 | 11.5 | 0.1 | 1.7 | -9.4 | 9.6 |
| C | Disc | 6.9 | 7.1 | -4.6 | 5.0 | -12.5 | 12.6 |
| | Cylinder | -2.7 | 3.0 | -4.9 | 5.1 | -11.9 | 12.0 |
| S | Disc | 5.6 | 5.9 | 5.3 | 5.5 | -15.2 | 15.3 |
| | Cylinder | 5.9 | 6.1 | 5.5 | 5.7 | -11.8 | 12 |
| L | Disc | -2.1 | 2.4 | 4.0 | 4.2 | -20.5 | 20.5 |
| | Cylinder | -1.8 | 2.2 | 4.1 | 4.3 | -20.5 | 20.5 |

Figure 8 shows that the cylinder parameterization performs better than the disc parameterization in $\sigma^0_{HH}$ simulation at the C-band, and the disc parameterization overestimates $\sigma^0_{HH}$ mainly due to the overestimated $\sigma^0_s$ (Fig. S17), while these two different parameterizations show the same performance in $\sigma^0_{VV}$ and $\sigma^0_{VH}$ simulations. The cylinder-based simulated $\sigma^0_{HH}$ at the C-band is close to the observation (RMSE of 3.0 dB in Table 5), but $\sigma^0_{VV}$ at the C-band is underestimated (Bias of -2.7 dB in Table 5) and

the simulated $\sigma^0_{VV}$ does not exhibit diurnal variations as the observation does (Fig. 8).

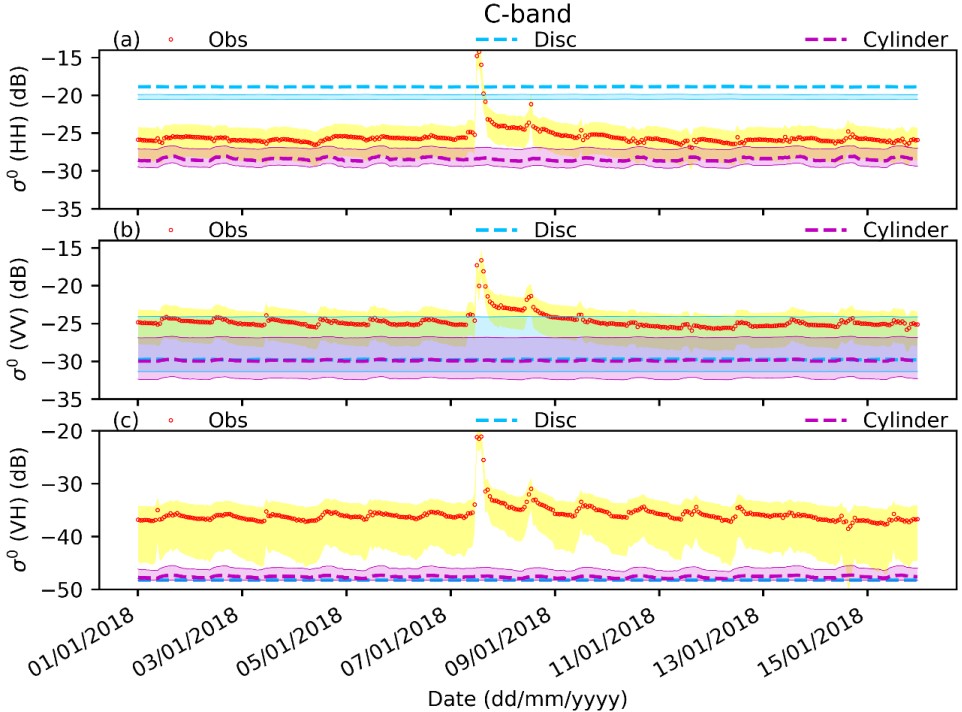

*Figure 8 Similar to Figure 7 but for C-band. The plot of in situ soil moisture and temperature at different depths and environmental variable observations in this period can be found in Figures 7a and 7b.*





Figures 9 and 10 show that the different vegetation structure parameterizations do not affect the

simulation of $\sigma_{pq}^0$ at the S- and L-bands during the winter period (small difference of 0.2 dB in Biases and

RMSEs at the S- and L-bands, respectively, in Table 5), and the soil contribution is dominant (Figs. S18

and S19). The simulated co-polarization $\sigma_{pq}^0$ at the S-band reflects the observed diurnal variations (Fig. 8

and Fig. S18), while simulated those at the L-band do not (Fig. 13 and Fig. S19). Figure 4 shows that the

$\sigma_{VH}^0$ at the X-, C-, S- and L-bands are all heavily underestimated (Biases < 0 for VH and RMSEs > 9.0 dB

in Table 5) during the winter period, especially at the L-band (RMSE of 20.5 dB in Table 5). This should

be due to the deficiency of the currently used AIEM model that does not involve the volume and multiple

scattering terms, whereas the volume scattering effect does present in the soil due to the presence of ice

and snow during this period (Figs. 7a and 7b).

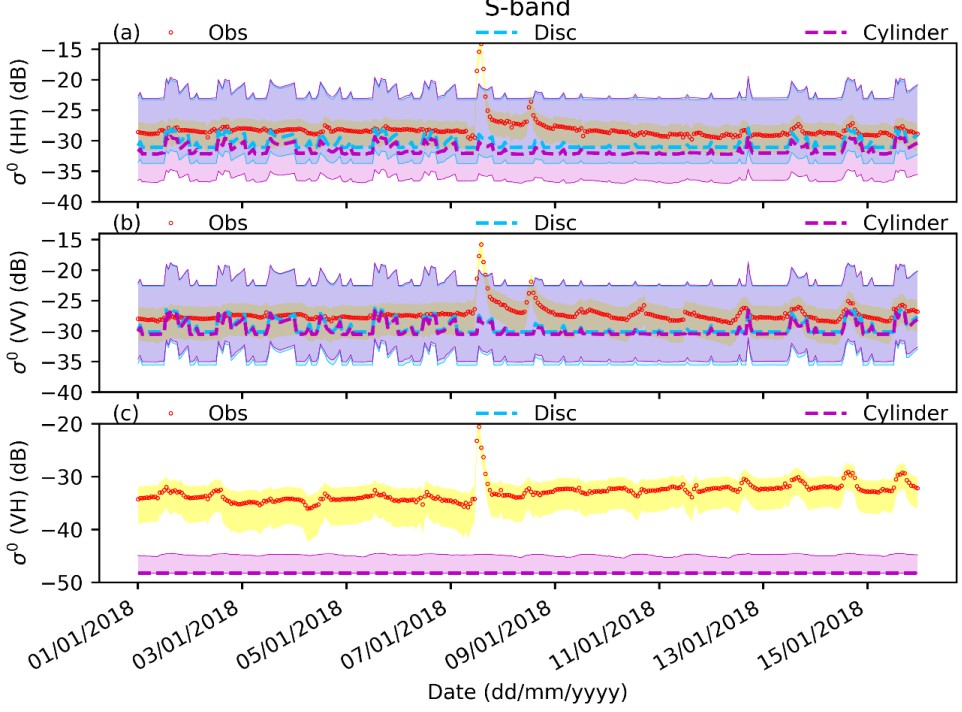

*Figure 9 Same as Figure 7 but for S-band.*





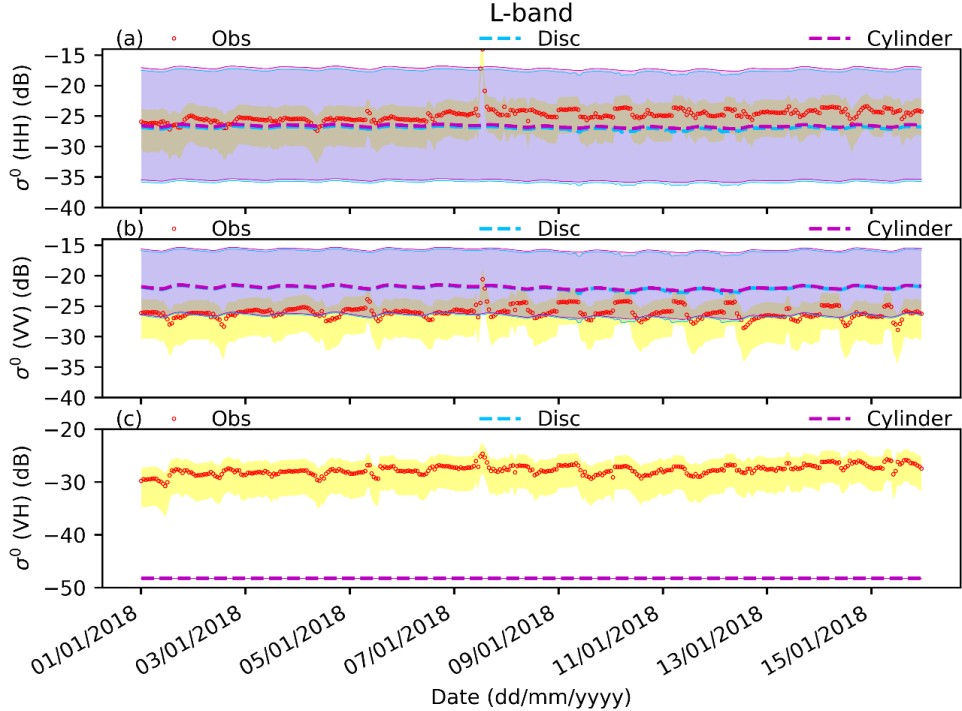

*Figure 10 Same as Figure 7 but for L-band.*

Next, we explore the impact of soil moisture on simulated backscatter using soil moisture at a shallower depth (e.g., 1 mm) than the *in situ* observed soil moisture at 2.5 cm. For this purpose, the soil moisture

and temperature profiles simulated by STEMMUS were used. Figure 11 shows that using STEMMUS simulated soil moistures and temperature as the input in the CLAP does not outperform using the *in situ* measured soil moisture and temperature in reproducing the observed $\sigma^0_{pq}$ at the X-band, and both lead to the heavy overestimation of $\sigma^0_{HH}$ at the X-band (Biases > 10 dB in Table 5). However, using STEMMUS simulated soil moisture and temperature outperforms in reproducing the observed $\sigma^0_{pq}$ at the C-band (Fig.

12). Moreover, the observed diurnal variation of $\sigma^0_{pq}$ at the C-band can be captured by the simulation (Fig. 12), which is not seen in the simulation results using the *in situ* soil moisture measured at 2.5 cm. This finding also applies to S-band at the HH polarization (Fig. 13), where the difference between $\sigma^0_{HH}$ simulated with the STEMMUS simulated soil moisture and temperature and the observation is lower than that of $\sigma^0_{HH}$ simulated with the *in situ* soil states against the observation (RMSEs of 2.4 dB vs 3.7 dB in

Table 5). While this improvement is not observed in simulating $\sigma^0_{VV}$, where using STEMMUS simulated soil states overestimates $\sigma^0_{VV}$, and the degree of overestimation is larger than that of the underestimation using the *in situ* soil states (Fig. 13, Biases of 4.6 dB vs -2.5 dB in Table 5).

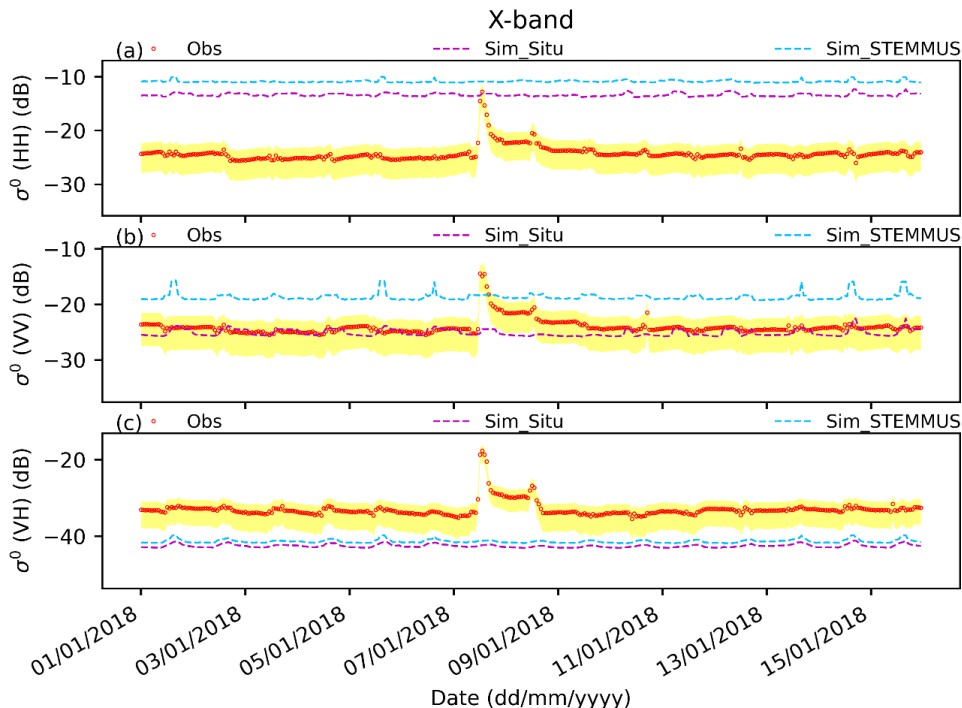

*Figure 11 $\sigma_{pq}^0$ at the X-band simulated by the CLAP (ATS_AIEM_TVG model) with the use of the in situ*
*measured and STEMMUS simulated profile soil moisture and temperature respectively, compared to the*
*scatterometer observations during the winter period. The yellow shaded area refers to the uncertainty in*





*the observed backscatter data (see Hofste et al. (2021)).*

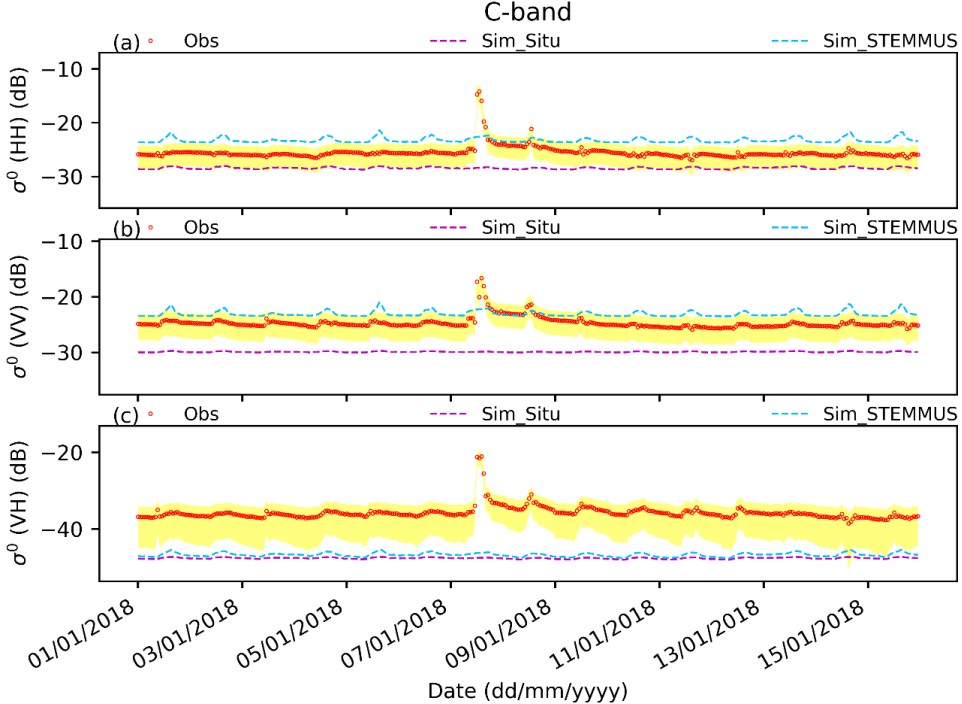

*Figure 12 Same as Figure 11 but for C-band.*


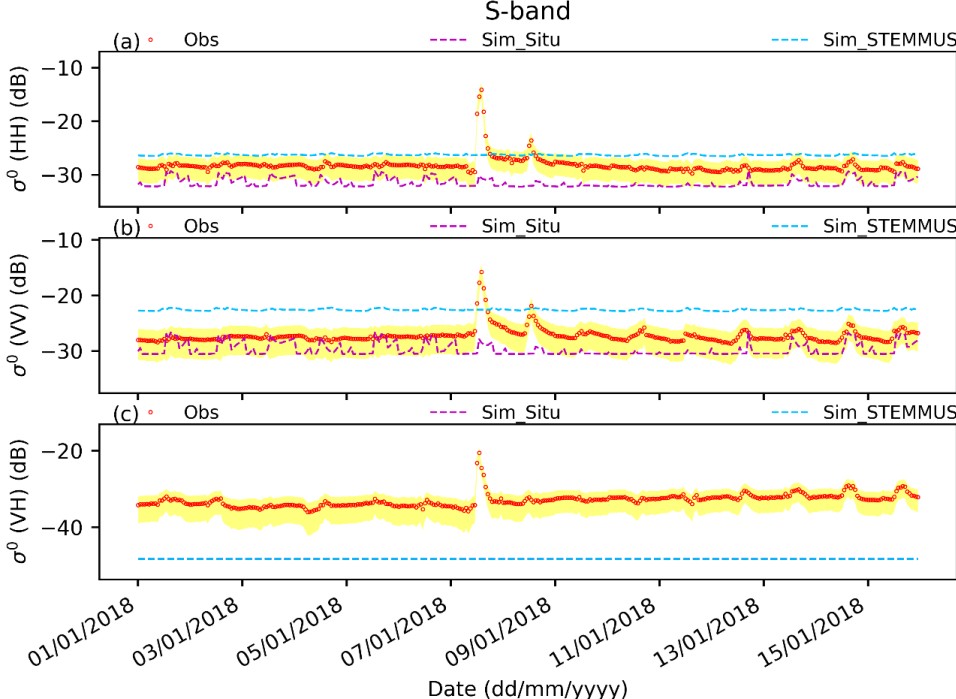

*Figure 13 Same as Figure 11 but for S-band.*

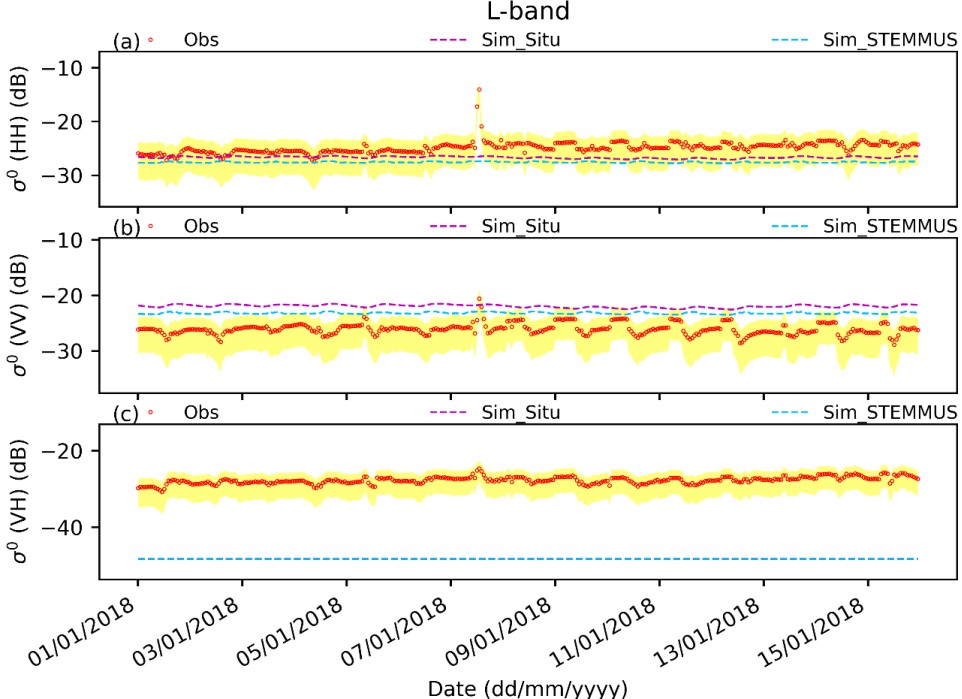

*Figure 14 Same as Figure 11 but for L-band.*

Both simulated co-polarization $\sigma_{pq}^0$ at the S-band shows diurnal variations that agree with the observations (Fig. 13). These analyses imply the different sensing depths of soil moisture through the observed backscatter at the different frequencies. Thus, obtaining precise dynamic surface soil (moisture) information and further investigating the foregoing sensing depth play a significant role in reproducing the observed diurnal microwave signal at high frequencies. Figure 14 shows a small difference in $\sigma_{pq}^0$ at

the L-band simulated with the *in situ* measured and STEMMUS simulated soil moisture and temperature (e.g., RMSEs of 4.3 dB and 3.1 dB under VV polarization in Table 6). This implies that not only the surface soil but also the soil at the deeper depth (e.g., soil temperature penetration depth in this case) contributes to the observed L-band signal since its stronger penetration capability than that at the S- and C-bands.

*Table 6 Statistics of the comparison of $\sigma_{pq}^0$ at multi-frequency simulated by the CLAP using the in situ measured and STEMMUS simulated profile soil moisture and temperature as the input respectively, to the ground-based observations during the winter period. Bias and RMSE are in the unit of dB.*

| Band | Scheme | HH | | VV | | VH | |
|------|--------|------|------|------|------|------|------|
| | | Bias | RMSE | Bias | RMSE | Bias | RMSE |



| | | | | | | | |
|---|---|---|---|---|---|---|---|
| X | *In situ* | 10.9 | 11 | -1.1 | 1.9 | -9.6 | 9.8 |
| | STEMMUS | 13.4 | 13.5 | 5.4 | 5.6 | -8.4 | 8.6 |
| C | *In situ* | -2.9 | 3.2 | -5.2 | 5.3 | -11.9 | 12.0 |
| | STEMMUS | 2.2 | 2.5 | 1.6 | 1.9 | -11.0 | 11.1 |
| S | *In situ* | -3.3 | 3.7 | -2.5 | 3.0 | -15.2 | 15.3 |
| | STEMMUS | 2.0 | 2.4 | 4.6 | 4.8 | -15.2 | 15.3 |
| L | *In situ* | -1.8 | 2.1 | 4.1 | 4.3 | -20.5 | 20.5 |
| | STEMMUS | -2.7 | 2.9 | 2.9 | 3.1 | -20.5 | 20.5 |

In short, the observed co-polar $\sigma_{pq}^0$ and its diurnal variations especially at VV polarization during the winter period can be reproduced by the CLAP using the process model (i.e., STEMMUS in this case)

simulated soil moisture and temperature as the input. However, the accuracy of soil moisture at different depths is important for good simulations of co-polar $\sigma_{pq}^0$ at multi-frequency. The observed cross-polar $\sigma_{pq}^0$ during the winter period cannot be reproduced well (RMSEs > 8 dB in Table 5) by the CLAP, especially at the longer wavelength (i.e., RMSEs > 14 dB at the S- and L-bands in Table 5).

### 3.4 Simulated emission ($T_B$) at multi-frequency during the winter period

Figure 15 and Figures S20 and S21 show that the variation of the ELBARA-III and AMSR2 observed $T_B^p$ signals at the L-, C- and X-bands, respectively, can be captured by the CLAP during the winter period. The cylinder parameterization outperforms in simulating $T_B^H$ at the X-band and $T_B^V$ at the C-band than the disc parameterization (Figs. S20 and S21). Due to the strong penetration capability of the L-band signal and the assumed dead dry vegetation during the winter period, the simulated $T_B^p$ at the L-band does not

differ for different vegetation parameterizations as shown in Fig. 15. Figure 15 also shows that the model can capture the observed diurnal variations well, although the large systematic underpredictions of $T_B^p$ are observed in comparison to the ELBARA-III observations (RMSEs of 26.3 K under V polarization and 48.9 K under H polarization in Table 7). This underestimation might be due to the deep modelled average dielectric surface (with the thickness of [h/2, h/2-log(SM1)* $s$], where h refers to the dielectric roughness

height and SM1 refers to the soil moisture of the first layer) used in the calculation of the effective dielectric constant of the composite air-soil medium in the ATS model (please refer to section 4.2).





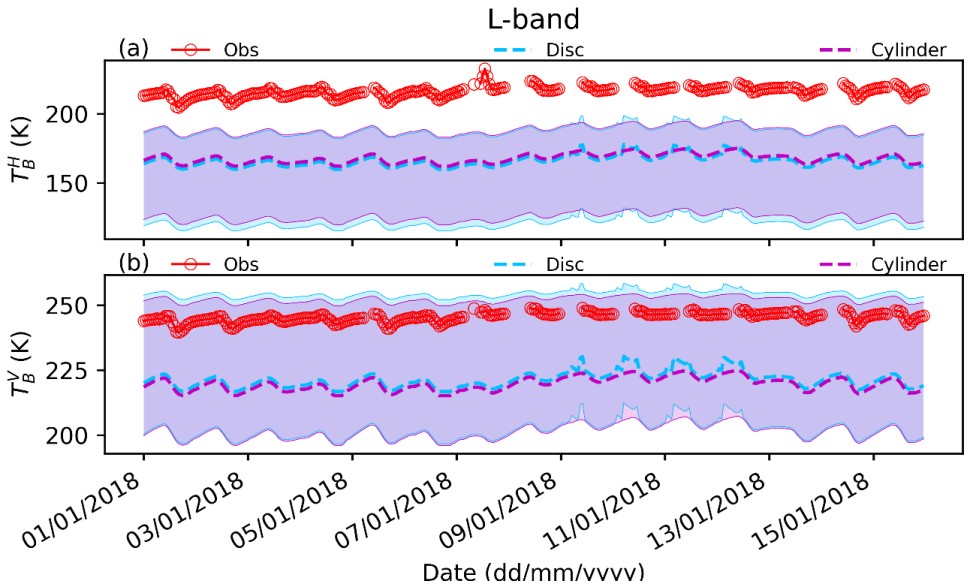

Figure 15 $T_B^p$ at the L-band simulated by the CLAP with the disc and cylinder parameterizations compared to the ELBARA-III observations during the winter period.

Table 7 Same as Table 4 but for the comparison of $T_B^p$ during the winter period. Bias and RMSE are in the unit of K.

| Band | Scheme | H | | V | |
|---|---|---|---|---|---|
| | | Bias | RMSE | Bias | RMSE |
| X | Disc | -40.4 | 43.2 | -14.3 | 8.5 |
| | Cylinder | -21.2 | 18.9 | 16.4 | 5.7 |
| C | Disc | -23.4 | 18.2 | 15.7 | 1.8 |
| | Cylinder | -15.8 | 11.1 | 15.8 | 1.8 |
| L | Disc | -51.2 | 51.3 | -24.7 | 24.7 |
| | Cylinder | -48.9 | 48.9 | -26.2 | 26.3 |

### 3.5 Comparison of estimated effective scattering coefficient ($\omega$) and optical depth ($\tau$)

Figure 16a shows that during the summer period, the estimated $\omega$ at each frequency based on the disc parameterization does not differ between HH and VV polarizations. The values of simulated $\omega$ at the C-

band are higher than those at the X- and S-bands, despite all of them being higher than 0.05 (Fig. 16a). In contrast, the estimated $\omega$ at the S-, C- and X-bands based on the cylinder parameterization exhibits values ranging from 0.02 to 0.04 and varies between different polarizations, in which the values of simulated Cylinder_$\omega$ at HH polarization are higher than those at VV polarization (Fig. 16c). It is noted that a global constant value of 0.06 was used in the X-band microwave emission of the biosphere model (Wang





et al., 2021a), while 0.08 was derived from the disc parameterization and 0.04 for the cylinder
parameterization in our simulation. The estimated $\omega$ at the L-band does not differ between different
polarizations, and the disc parameterization produces  mean value of 0.02 of $\omega$, which are higher than
zero values of $\omega$ produced by the cylinder parameterization (Figs. 16a and 16c), but lower than 0.05 used
by the SMAP soil moisture retrieval algorithm for grassland as reported by Zheng et al. (2018b). Figures

16a and 16c show that the $\omega$ is suppressed by rainfall events during the period from 01-08-2018 to 03-08-
2018 (see Figs. 2a and 2b), and this is because the vegetation temperature assigned from air temperature
drops and then undergoes stable variations in this period, resulting in reduced vegetation emissivity and
therefore decreased $\omega$ estimated by Equation (2a). While the $\omega$ increases after the rainfall, which might be
due to the increased soil moisture as reported by Kurum (2013).

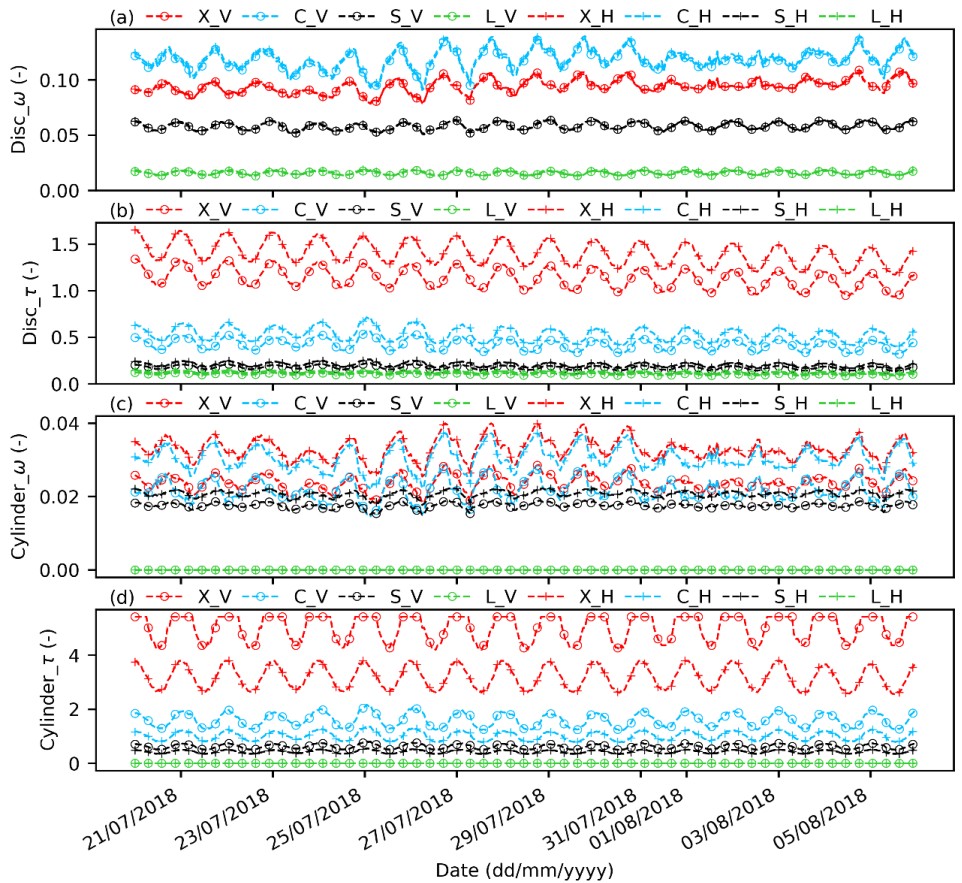


*Figure 16 Comparison of estimated $\tau$ and $w$ at the multi-frequency during the summer period by the
CLAP (ATS_AIEM_TVG model) using the disc and cylinder parameterizations respectively.*



Figure 16b shows that the disc parameterization produces $\tau$ at S-, C- and X-bands ranging from 0.2 to 1.7 during the summer period, and values of simulated $\tau - H$ at these bands are higher than those of the

corresponding simulated $\tau - V$. Comparably, the simulated Disc_$\tau$ at the L-band under both polarizations exhibits a low mean value of 0.1. The range of values of disc parameterization derived $\tau$ is close to the satellite $T_B^p$-derived $\tau$ reported by Li et al. (2021), and this is expected as the disc parameterization shows good performance in simulating satellite $T_B^p$ at the X- and C-bands (please refer to section 3.1). In contrast, the cylinder parameterization produces a large range of $\tau$ at multi-frequency and high values

ranging from 0 at the L-band to 6.0 at the X-band (Figs. 16b and 16d). As the simulated microwave signal based on cylinder parameterization matches the *in situ* observations as shown in section 3.1, we conclude that the estimated Cylinder_$\tau$ is valid. The values of the simulated Cylinder_$\tau$ at the X-, C- and S-bands under VV polarization are higher than those under HH polarization, while the simulated Cylinder_$\tau$ at the L-band is polarization independent.

Figures 17a and 17b show that the disc parameterization produces zero values of $\omega$ and $\tau$ at the C-, S- and L-bands under both polarizations, while high values of $\omega$ around 0.06 and low values of $\tau$ around 0.001 at the X-band under both polarizations. This indicates that the vegetation during the winter period exhibits weak scattering at the X-band but becomes transparent at the C-, S- and L-bands. In contrast, the cylinder parameterization produces lower values of $\omega$ ranging from 0.02 to 0.03, and higher values of $\tau$ ranging

from 0.08 to 0.15 at the X-band under both polarizations, and non-zero values of $\omega$ around 0.004 and $\tau$ around 0.04 at the C- and S-bands as well (Figs. 17c and 17d). Similar to those based on the disc parameterization, the estimated Cylinder_$\omega$ and Cylinder_$\tau$ at the L-band also exhibit zero values. Figures 17c and 17d show that the cylinder parameterization produces higher values of the simulated $\tau$ at the X-, C- and S-bands under HH polarization than those under VV polarization, and this is reasonable as

the horizontal orientation of vegetation during the winter period is assumed in this study.

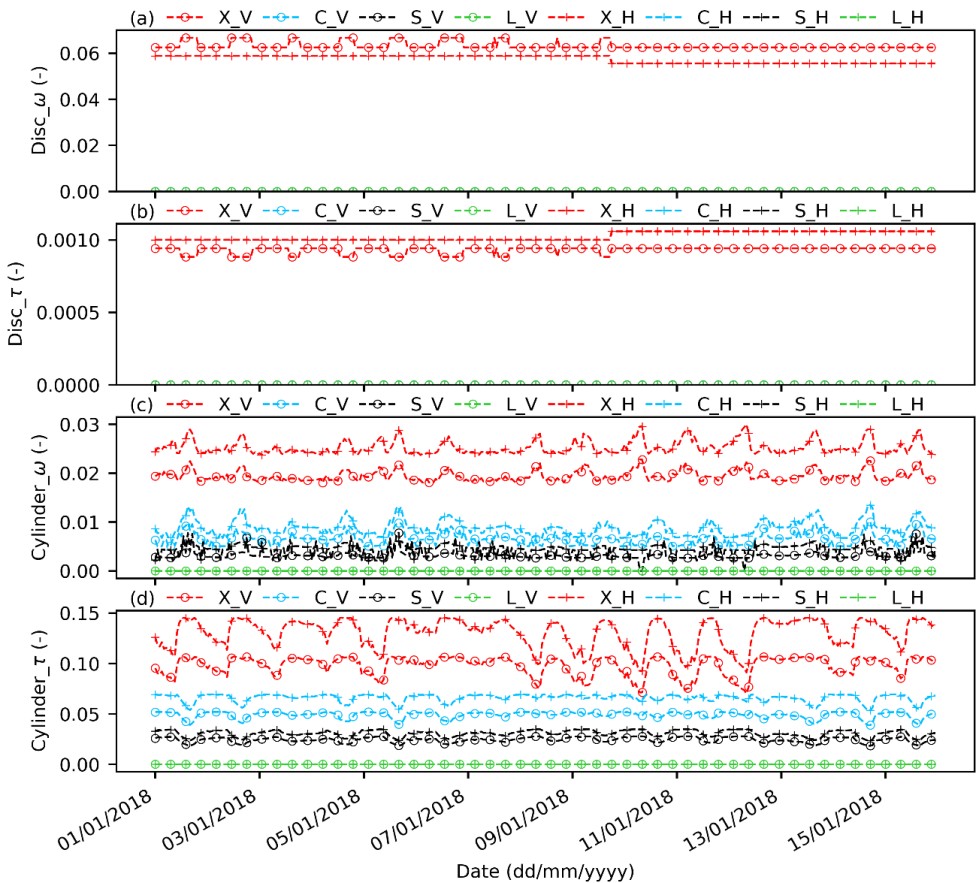

*Figure 17 Same as Figure 16 but for the winter period.*

All these analyses imply that the ω of grassland at multi-frequency depends on the plant status (e.g., live with high water content or senescent with low water content) and structure. The grass structure

determines the polarization-dependent behavior mainly at the high frequencies (i.e., X-, C- and S-bands), while it does not show significant impacts on differing estimated L-band ω and $\tau$ at the different polarizations. The diurnal cycles of the estimated ω and $\tau$ (Figs. 16 and 17) are due to the consideration of the dynamic vegetation temperature and water content on the diurnal scale in the CLAP modelling, which is consistent with those reported by Vermunt et al. (2021) and Humphrey and Frankenberg (2022).





## 4. Discussion

### 4.1 Sensitivity of microwave $\sigma_{pq}^0$ simulation to vegetation orientation during the winter period

Since the totally horizontal orientation of vegetation (beta angle of 90° in Table 2) is assumed during the winter period in this study, it decreases the contribution (e.g., multiple reflections between vegetation and soil) to the cross-polarization $\sigma_{pq}^0$, and as a result the observed cross-polarization $\sigma_{pq}^0$ at high frequencies (X-, C- and S-bands) during this period is underestimated by the CLAP (Figs. 7-14). The beta angle (Table 2) ranging from 60° to 90° is set in the *in situ*-based simulation to investigate the impact of different distributions of leaf angle on $\sigma_{pq}^0$ simulation. The results (Figs. S22 and S23) show that the different leaf angle orientation does not affect the simulated co-polarization $\sigma_{pq}^0$ at the X- and C-bands, while it improves the model performance in the simulation of cross-polarization $\sigma_{pq}^0$. The improvement is also seen in the simulation of $\sigma_{pq}^0$ at the S-band (Fig. S24) but not at the L-band (Fig. S25). The heavy underestimation (Bias of -20.5 dB in Table 5) of cross-polar $\sigma_{pq}^0$ at the L-band may be due to the volume and multiple scattering terms missing in the AIEM model.

### 4.2 Improving the simulation of $T_B^p$ at the L-band during the winter period

In the default ATS model, there are two critical parameters: soil moisture and the dielectric thickness of the boundary condition, with the latter parameter used in the averaging procedure for calculating the effective dielectric constant of the composite air-soil medium (Zhao et al., 2021). The default parameterization that uses soil moisture at 2.5 cm (SM1 in Table 8) and the deep boundary b1 = [h/2, h/2-log(SM1)* $s$] (Case0 in Table 8 and Fig. 18), leads to a large systematic underprediction of $T_B^p$ at the L-band in comparison to the ELBARA-III observations as shown in Fig. 15. To investigate the impact of these two parameters and improve the $T_B^p$ simulation during the soil freeze-thaw period, the extra three simulation experiments for ten days are carried out. As described in Table 8, the first experiment (Case1) uses SM1 the same as Case0 as the input, but assumes a shallow average dielectric surface with a thickness of [0, -log(SM1)*$s$] implying that the signal variation is more due to the surface soil freeze-thaw process. The second experiment (Case2) continues to use the b2 condition but with soil moisture at the penetration depth (SM_pd) of soil temperature (Lv et al., 2018). This configuration considers both surface condition and soil condition at the penetration layer. The third experiment (Case3) uses SM_pd and a new boundary b3 = [wavelength/10,-log(SM_pd)*wavelength/10]. This configuration considers also both soil condition at the penetration layer and the surface condition, but with the latter reflected along the theoretical effective penetration depth of soil moisture (i.e., 1/10 of the wavelength as described in section 1).





*Table 8 Configurations of ATS simulation experiments.*

| Variable | Case0 | Case1 | Case2 | Case3 |
|---|---|---|---|---|
| SM (m³/m³) | SM1 | SM1 | SM_pd | SM_pd |
| The thickness of the average dielectric surface (cm) | b1 = [h/2, h/2-log(SM1)*s] | b2 = [0, -log(SM1)*s] | b2 | b3 = [wavelength/10, -log(SM_pd)* wavelength /10] |

*where SM1 denotes soil moisture at the first layer, SM_pd denotes soil moisture at the penetration depth of soil temperature. h denotes the dielectric roughness height. The definition of h and average dielectric surface thickness (b) can be found in Zhao et al. (2021).*

Figure 18 shows that the Case1 can reproduce the observed diurnal variation of $T_B^p$ signals well but present a heavy systematic overprediction (RMSEs of 26 K at V polarization and 48 K at H polarization in Table S3) of $T_B^p$, which is opposite to the underprediction by Case0. The values of the observed $T_B^p$ seem bounded by the simulation results from Case0 and Case1. A phase delay is observed between the $T_B^p$ simulated by both Case0 and Case1 and the observation (Fig. 18). The phase delay is understandable,

since the ELBARA-III $T_B^p$ observations reflect the real surface condition, while the used model inputs of soil moisture and temperature measured at 2.5 cm are the delayed information driven by solar radiation and soil heat and water transport processes. Regarding the active case, Case 1 that utilizes the surface information can generally reproduce the observed diurnal variations of co-polarization $\sigma_{pq}^0$ (Fig. S26). In contrast, Case2 (using soil moisture at the penetration depth and the surface shallow boundary condition)

captures the magnitude of the observed $T_B^H$ (RMSE of 5 K in Table S3) and $\sigma_{VV}^0$ (RMSE of 0.9 dB in Table S4), Case 3 (using soil moisture at the penetration depth and wavelength information) mimics the magnitude of the observed $T_B^V$ (RMSE of 5 K in Table S3) and $\sigma_{HH}^0$ (RMSE of 3.3 dB in Table S4), although the diurnal changes simulated by both cases are flat. These comparison results indicate that the passive H-polarization signal reflects the surface condition and the passive V-polarization signal reflects

more about the soil states at the penetration depth, which is the other way around in the active case. The parameterization as such is sufficient to model the contribution of soil at the penetration depth, while the surface layer imposing its impact through the freeze-thaw condition needs to be investigated further with the aid of accurate information of surface soil states, which is difficult to obtain (either through *in situ* or the process model).

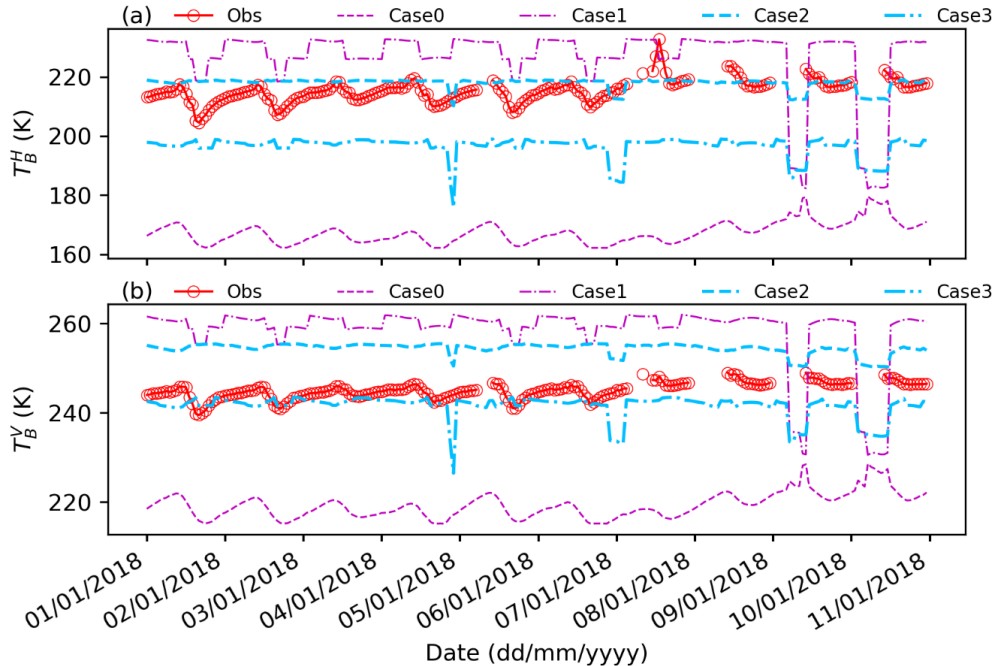


*Figure 18 Comparison of $T_B^p$ at the L-band estimated by four experimental cases to the ELBARA-III*

*observations during the winter period. Case0 refers to the parameterization using soil moisture at 2.5 cm*

*(SM1) and the average dielectric surface with thickness b1 = [h/2, h/2-log(SM1)\* s] in the ATS model.*

*Case1 refers to the parameterization using SM1 the same as Case0 and the average dielectric surface*

*with thickness b2 = [0, -log(SM1)\*s] that represents the shallow surface condition. Case2 refers to the*

*parameterization using the b2 condition and soil moisture at the penetration depth (SM_pd) of soil*

*temperature (Lv et al., 2018). Case3 refers to the parameterization using SM_pd and a new boundary of*

*b3 = [wavelength/10, -log(SM_pd)\* wavelength/10] that considers the wavelength information and the*

*surface condition along the theoretical effective penetration depth of soil moisture (i.e., 1/10 of the*

*wavelength as described in section 1).*

On the other hand, the abrupt jump of the observed co-polarization $\sigma_{pq}^0$ (on 07/01/2018 in Figs. 7-10) and

$T_B^p$ (Fig. 18) due to the snowfall event (please refer to the high albedo value in Fig. 7a) cannot be captured

by the current setup, as no snow information is involved in the modelling, for which future investigation

should be conducted. The similar jumps of the observed $\sigma_{pq}^0$ and $T_B^p$ (see Figs. 2-6) are also observed due

to the rainfall event. Figure 2 shows that the effect of a light rainfall (< 2.0 mm/hour) on the observed

signal is pronounced at the X-band at full polarizations, and at the C- and S-bands at cross-polarization





(see Figs. 3 and 4). The impact of heavy rainfall (> 6.0 mm/hour, Fig. 2a) on the observed signal is pronounced at the X- and C-bands at full polarizations, at the S-band at cross-polarization, and at the L-band at HH- and cross-polarizations (see Figs. 2-5). This finding is consistent with those reported by

Vermunt et al. (2021) and Khabbazan et al. (2022), in which the radar backscatter of corn fields at the L-band is found exhibiting diurnal variations partially due to the intercepted water. As there is also no *in situ* measured interception water in this study, the current setup cannot capture the foregoing phenomenon. It is, however, to note that the anomaly of the systematic difference between the observation and the current CLAP simulation results could be utilized to detect snow and rainfall events.

Additionally, a future process-based investigation can be conducted based on the modelled interception water with the land surface model.

### 4.3 Normalization of estimated optical depth ($\tau$)

Different $\tau$ can be obtained by using different frequency microwave signals and different model algorithms, which may become inconvenient for comparison. To link $\tau$ from different frequencies, we

proposed to use wavenumber $k_0$ ($\frac{2\pi}{wavelenth}$) to normalize the estimated $\tau$. The results shown in Fig. 19 indicate that the normalized $\tau$ ($\tau/k_0$) does not converge to one value and still depends on frequency, implying that $\tau$ is indeed frequency-dependent, and the retrieved $\tau$ at different frequencies may reflect either different vegetation layers or different properties of vegetation . As such, $\tau$ at different frequencies cannot be directly intercompared and $\tau$ for each frequency is suggested to be applied in the retrieval of

soil and vegetation parameters . The disc-based $\tau/k_0$ for each frequency shows a decreasing trend (Fig. 19), which is due to the number of discs that is related to LAI (see Table 2 and Figure 2a). While the cylinder-based $\tau/k_0$ for each frequency does not show a trend (Fig. 19) as the number of cylinders is set constant in this case (see Table 2), which is noted being constrained indirectly via the measured fresh biomass (Hofste et al., 2022). Figure 19 also shows that the diurnal shape of $\tau$ is determined by the shape

of the estimated VWC regardless of frequency. This finding further demonstrates the sensitivity of the microwave derived $\tau$ to both plant biomass and water content, and supports the rationality of current research using radar backscatter to detect dynamic VWC (Steele-Dunne et al., 2017), while the results shown in our study implies the need to first disentangle the impact of vegetation temperature for high frequency signals, since the diurnal variation of the observed signals at high frequencies is found more

due to vegetation temperature changing (see section 3.1 and Fig. S27) that is reflected on estimated $\omega$ (see Fig. 16). As such, the retrieved $\tau$ at different frequencies (e.g., X-band and L-band) cannot be simply combined for constructing the long-term global microwave-based vegetation product. Instead, the radiative transfer modelling approach needs to be considered for combination, but this is beyond the scope of this study. While as leaf water potential, which measures plant water status, is the driving force



behind VWC (Konings et al., 2019). The difference in $\tau$ at multi-frequency even after the normalization
       might be due to the difference in leaf water potential, which is sensed differently by varied frequency
       signals, but this aspect needs to be further investigated by, either combining CLAP with the process
       model or using the measured leaf water potential data, which is unfortunately not available in this study.
       Using microwave signals to probe leaf water potential can help understand plant water use regulation and

the associated drought vulnerability on the global scale.

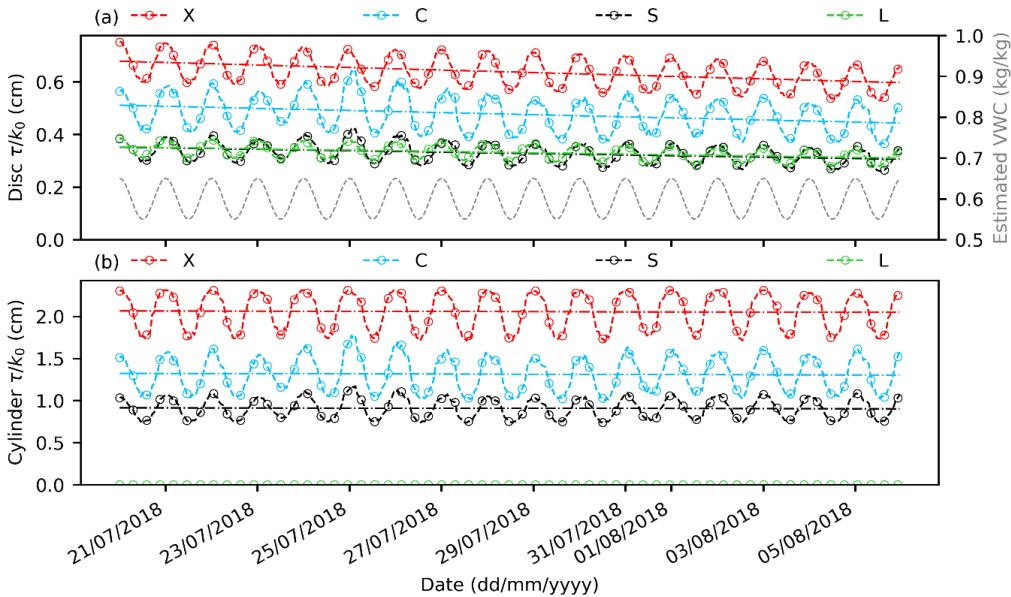

*Figure 19 Wavenumber ($k_0$) normalized optical depth ($\tau$) estimated by the disc and cylinder
parameterizations respectively, during the summer period. The dashed line denotes the trend.*

**4.4 Potential application of CLAP**

Although model performance from several aspects as discussed in sections 4.1 and 4.2 needs to be
       improved and the model validity for other vegetated lands needs to be further demonstrated, the CLAP
       has been proven in this study as a key tool to advance our knowledge in understanding the microwave
       scattering-emission mechanism of vegetated lands by simulating simultaneous active and passive
       microwave observations at the Maqu observatory. It is noted that the essential CLAP inputs such as

moisture content and temperature of soil and vegetation as well as vegetation geometry (e.g., size, shape,
       orientation and distributions of elements) are not always available *in situ*. One of the innovative
       approaches to circumvent this limitation is to include the integrated process model STEMMUS-SCOPE
       (Wang et al., 2021b) in the CLAP, in which the SCOPE (Van Der Tol et al., 2009) modelling canopy





radiative and photochemical processes provides vegetation information and STEMMUS model (Zeng et
al., 2011a; Yu et al., 2018; Yu, 2022) provides information of profile soil moisture and temperature. The
corresponding investigations carried out with this new integrated framework are expected to provide
insights for resolving the challenge of remote sensing of vegetation: to be able to describe and separate
the contributions of the different components in the observed total signature of the vegetated lands, to
further root satellite observations into the actual surface conditions for ecological and hydrological
applications. The role of leaf water potential described in section 4.3 can also be investigated by using
this framework.

Consequently, three main applications of CLAP are feasible. The first application would be to simulate
observations from current and future multi-frequency space-borne microwave systems (e.g. ROSE-L and
CIMR), derive the added value (e.g., a physically consistent dynamic $\tau$ and $\omega$ dataset) and test the
operational soil moisture and vegetation parameter retrieval algorithms. The second application would be
to include CLAP as an observation operator in the data assimilation framework, which is useful for
estimating vegetation and soil  properties and land surface fluxes with land surface models in a physically
consistent manner. The third application is to utilize CLAP to conduct sensitivity studies to explore
physical meaningful parameter space for the use of machine learning in soil moisture and vegetation
parameter estimation (Stamenkovic et al., 2017; Gao et al., 2022; Chaudhary et al., 2022).

## 5. Conclusions

In this study, we describe a Community Land Active Passive Microwave Radiative Transfer Modelling
Platform (CLAP) that can be used for integrated modelling, interpretation and application of multi-
frequency emission and backscattering signals of land surfaces. Specifically, the CLAP is backboned by
an air-to-soil transition model (ATS) (accounting for surface dielectric roughness) integrated with the
Advanced Integral Equation Model (AIEM) for modelling soil surface scattering, and the Tor Vergata
model for modelling vegetation scattering and the interaction between vegetation and soil parts. In
comparison to the *in situ* and satellite microwave observations at the Maqu site on the Eastern Tibetan
Plateau, the CLAP has been demonstrated to be capable to reproduce the observed multi-frequency
microwave backscatter $\sigma_{pq}^0$ and emission $T_B^p$ signals of grassland during the summer period well, and the
CLAP using the cylinder parameterization of vegetation representation can mimic multi-frequency $\sigma_{pq}^0$
better than the disc parameterization does. Regarding the diurnal variation of the observed signal at the
high frequencies (i.e., X- and C- bands), the simulation comparison results indicate that dynamic
vegetation temperature partially accounts for, while the dynamic VWC partially results in the diurnal
variation of the observed signal at the low frequencies (i.e., S- and L-bands). Accordingly, the CLAP-





derived effective scattering coefficients ω and vegetation optical depth (VOD or τ) exhibit diurnal variations, which are due to the impact of the dynamic vegetation temperature and water content on the vegetation dielectric constant.

The CLAP using the cylinder parameterization and either the *in situ* measurements or the process model outputs can mimic the observed co-polarization $\sigma^0$ of grassland and its diurnal variations especially at VV polarization during the winter period, in which the accuracy of soil moisture at the penetration depth influences the good simulation of co-polarization $\sigma^0$ and $T_B^p$, and the vegetation orientation information plays an important role in obtaining good simulations of cross-polarization $\sigma^0$ at high frequencies (i.e., X- and C-bands). However, the current platform cannot mimic the observed cross-polarization $\sigma^0$ at the L-band during the winter period, which might be due to the deficiency of the currently used AIEM model that does not involve volume and multiple scattering terms.

Future work may apply CLAP modelling for agricultural plants, for instance, the maize field in the Reusel site in the Netherlands (Vermunt et al., 2021), where the satellite microwave observations (e.g., Sentinel-1 C-band backscatter, SAOCOM X-band and L-band backscatters, and AMSR2 and SMAP $T_B^p$) and the ground-based active observations (X, C- and L-bands, per communication with Susan C. Steele-Dunne), and the *in situ* measured vegetation parameters and soil moisture and temperature profiles are available for validation. On the other hand, CLAP will be extended as a full-spectrum observation operator by adding SCOPE modelling canopy radiative and photochemical processes and STEMMUS modelling soil water, heat and vapor transfer processes modeled by model, which can be used to synergistically utilize available and future satellite resources for monitoring (vegetation and soil) variables of interest at the global scale.

## Author contribution

ZS, HZ, YZ and JH developed the research idea, objectives and methodology. HZ and JH prepared datasets. HZ conducted simulations, applied the analysis and wrote the draft. TD contributed to the interpretation of related results. JW maintained the Maqu network and helped with data collection. HZ, ZS, YZ and JH revised the initial draft of the manuscript.

## Competing interests

The authors declare that the co-author Bob (Zhongbo) Su is the editor in HESS journal.



## Acknowledgments

770 The authors would like to thank European Space Agency (ESA) for providing the ELBARA-III radiometer. This work is carried out under the MINERVA: MIcrowaves for a New Era of Remote sensing of Vegetation for Agricultural monitoring project funded by the Netherlands Organization for Scientific Research (NWO) Partnerships for Space Instruments & Applications Preparatory Programme (PIPP) (KNW19001).

775

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
