# Peer review of "Modelling of Multi-Frequency Microwave Backscatter and Emission of Land Surface by a Community Land Active Passive Microwave Radiative Transfer Modelling Platform (CLAP)"

_Hydrology and Earth System Sciences, 2022_

## Referee Comment (RC1)

**Review of "Modelling of Multi-Frequency Microwave Backscatter and Emission of Land Surface by a Community Land Active Passive Microwave Radiative Transfer Modelling Platform (CLAP), by Zhao et al.**

*Reviewer: David Chaparro*

This paper presents a multifrequency active-passive transfer modelling, named CLAP, for simulating backscatter and emission signals of a vegetated (herbaceous) surface in the Maqu area in Tibet. Results are compared with ground-based radiometer and scatterometer measurements, and with AMSR2 data. The authors explore how simulations using different frequencies (L-, S-, C- and X-bands), vegetation configurations (cylinders vs. discs), seasons (winter vs. summer) and soil moisture sources (*in situ* vs modelled) track radiometer and scatterometer observations. With this, they aim at enhancing our understanding of vegetation, soil, and temperature impacts on daily variations in emission and backscatter.

The paper is well written and encompasses a wide range of characteristics influencing model outputs, thus being complete research. The work improves previous papers estimating microwave emissions by including the backscatter component in the outputs. Overall, it is research with good potential, but I have relevant major concerns that must be thoroughly addressed through the manuscript, especially regarding that: (i) the metrics used in the results should be extended, (ii) it is needed an extended discussion based on literature previously mentioned by the authors in the introduction, and (iii) the paper needs putting the results obtained in the context of the current and future missions and retrievals (e.g., which could be the impact of the reported RMSE and Biases in SM retrievals?). The third point is especially important, as the authors emphasize the applicability of the model stating that it can mimic observations and that it could be applied to global scales. None of these are fully justified if the reader cannot understand how CLAP could be applied: how can it be transferred to global scale? Which could be the impact of model errors if it was to be applied in moisture estimates?

I detail my comments hereafter.

**Major comments**

- The results presented are based on RMSE and Bias metrics between the estimates and the in situ, as well as on time-series plots. However, as the authors focus on the study of the daily variability, correlation metrics are needed. Also, optionally, scatters between *in situ* and estimates can be plotted next to the time-series (instead of in separate figures; note that the number of figures in the paper is very large and should not be increased).

- Lines 120-132, and 142-145, in p. 5, provide an interesting state of the art which is used as the basis for this paper. The manuscript will improve a lot if the authors extend the discussion explaining how the current manuscript improves previous literature. For instance (among others) the authors could address the question: "which are the improvements if compared to Zheng et al. (2021) and Dente et al. (2014)?"

- The authors conclude that the model is able to track well the observations in many cases (especially for cylinders and in summer). Still, the results show some differences between estimates and observations. In that sense:
  - In Figures 2 to 5 and Table 3, even in the best cases (cylinders and X to S frequencies) the errors reach RMSEs between 1 and 4. As the authors are presenting this model as potentially applicable for future missions, how would an error like this impact soil moisture and optical depth retrievals? Based on either literature or observations, the authors should discuss which is the impact of this error and if it is small enough to allow the applicability of the algorithm.
  - Table 4: similar to above. RMSE minimum values are 5.9 and 2.4 at H and V polarizations. Which would be the impact in soil moisture simulations? Is the conclusion of CLAP "mimicking the observations" consistent according to these results? Why? Maybe, the problem is that affirming that the model reproduces well the observations is subjective if there is no reference for what is "good" and what is "bad" (in terms of amount of error). Can the authors provide reference or thresholds of errors to justify why their affirmation of CLAP mimicking the observations is true enough to allow the model applicability?
  - Similar problems arise in Figures 7 to 14, but in this case they are well justified in the discussion.
  - Similarly, the following sentence would need justification answering the question: "why are the errors low enough to affirm that the CLAP is reproducing or close enough to observations?":
    - P. 29, l. 508-509: "In short, the observed co-polar $\sigma_{pq}$ and its diurnal variations especially at VV polarization during the winter period can be reproduced…".
  - L. 515-516: Figure 15 shows differences between disc/cylinder and observations of around 50K in H and 25K in V. Based on this, at L-band, the affirmation that the model is reproducing the observations is not true. Please review the sentence and derived conclusions through the paper.

- In some sentences, the authors derive conclusions or potential applicability at global scale, which is not demonstrated by the results. These sentences should be rephrased:
  - L. 686-687: even if it is shown in the results that combination of frequencies is not enough for constructing a homogeneous time-series of $\tau$, it cannot be concluded that this would happen globally. It could be said that this is the case for the study area and maybe in other grasslands.
  - L. 760-761: suggesting that CLAP can be applied at a global scale is maybe premature. The model has potential, but I suggest saying that larger scales (up to global) should be studied in the future. Let it open as a future work rather than an affirmation.

**Minor comments**

- Title: can you explain briefly in the introduction why the word "Community" is used as part of the name of the algorithm?

- L. 26: "simulate both ground-based and space-borne". This sentence may lead the reader to think that two different simulations (one for ground-based and the other for space-borne) are built. I think that the differentiation in these two types is more appropriate when you talk about the sensors used for validation.

- Abstract & introduction: I suggest being specific from the beginning of the paper stating that this analysis is conducted in soils covered by herbaceous vegetation.

- L. 45 – 47: following my major comments, reevaluate if this conclusion can be driven from the results. It should be further justified.

- L. 55: instrument → instrument**s**

- L. 59-60: maybe add the soil texture here?

- L. 75-77: in addition to the Steele-Dunne paper maybe you want to add further references referring specifically to agriculture. Some suggestions: Patton & Hornbuckle (2012), Hornbuckle et al. (2016), Chaparro et al. (2018), Mateo-Sanchís et al. (2019); Weiss et al. (2020).

- L. 87-89: with some exceptions, such as the Multi-temporal Dual Channel Algorithm (MT-DCA).

- L. 94-95: I cannot understand clearly what you mean with the first sentence of the paragraph. Please rephrase or remove.

- L. 141: after "Wang, 1987), a comma should follow instead of a full stop.

- L. 153: "temprature" → "temperature".

- L. 216: temperature → VWC

- L. 225-226: "the simulated data during the winter period is focused, as we assume…". Review the structure of the sentence.

- L. 226-227 & Fig. S2: at 2 cm, the simulated soil moisture fluctuates much more than the observed SM. Even if we do not have 1 mm observations, it is reasonable to think that the same "excess of fluctuations" could happen at the skin layer. How can this impact the results?

- Fig. 1: do you mean "Rayleigh-Jeans" instead of "Ryleigh-Gans"?

- Fig. 2b: the dashed line is not in the legend.

- Fig. 4b: the high variability of the observations between 31-7-18 and 5-8-18 is not captured by the model, which in general (also in other figures) shows a very regular,

sinusoidal behavior, not capturing extreme deviations such as these ones. Could you discuss why this happens and its implications, please?

- L. 457: Fig. 8 → Fig. 9

- L. 458: Fig. 13 → Fig. 10.

- L. 458: "simulated those" → "those simulated"

- L. 480 and 482: Table 5 → Table 6.

- L. 540: for further comparison of the albedo values, Baur et al. (2021) could be an interesting reference.

- L. 545: "suppressed". Instead, I would say "reduced".

- Figure 16: for readers who are used to see VOD ($\tau$) satellite retrievals which are not polarization dependent, values of up to 5 in $\tau$ for a grassland are absolutely out of the expected range. Is this expected in the vertical polarization? Why?

- L. 670: wavelenth → wavelength

- L. 689-690: is this sentence incomplete?

- L. 694-695: you could add the reference Jagdhuber et al. (2022) as an example, optionally.

- L. 702: in line with my major comments, affirming that CLAP has been proven as a key tool for understanding is too much if no impact of the errors in the potential retrievals is assessed.

- L. 706-708 and 758: SCOPE has not been presented or explained before in the paper. I suggest removing it or explaining previously.

- L. 717-725: review specifically this paragraph according to the major comments.

**References**

Baur, M. J., Jagdhuber, T., Feldman, A. F., Chaparro, D., Piles, M., & Entekhabi, D. (2021). Time-variations of zeroth-order vegetation absorption and scattering at L-band. Remote Sensing of Environment, 267, 112726.

Chaparro, D., Piles, M., Vall-Llossera, M., Camps, A., Konings, A. G., & Entekhabi, D. (2018). L-band vegetation optical depth seasonal metrics for crop yield assessment. Remote Sensing of Environment, 212, 249-259.

Hornbuckle, B. K., Patton, J. C., VanLoocke, A., Suyker, A. E., Roby, M. C., Walker, V. A., ... & Endacott, E. A. (2016). SMOS optical thickness changes in response to the growth and development of crops, crop management, and weather. Remote Sensing of Environment, 180, 320-333.

Jagdhuber, T., Jonard, F., Fluhrer, A., Chaparro, D., Baur, M. J., Meyer, T., & Piles, M. (2022). Toward estimation of seasonal water dynamics of winter wheat from ground-based L-band radiometry: a concept study. Biogeosciences, 19(8), 2273-2294.

Mateo-Sanchis, A., Piles, M., Muñoz-Marí, J., Adsuara, J. E., Pérez-Suay, A., & Camps-Valls, G. (2019). Synergistic integration of optical and microwave satellite data for crop yield estimation. Remote sensing of environment, 234, 111460.Patton, J., & Hornbuckle, B. (2012). Initial validation of SMOS vegetation optical thickness in Iowa. IEEE Geoscience and Remote Sensing Letters, 10(4), 647-651.

Patton, J., & Hornbuckle, B. (2012). Initial validation of SMOS vegetation optical thickness in Iowa. IEEE Geoscience and Remote Sensing Letters, 10(4), 647-651.

Weiss, M., Jacob, F., & Duveiller, G. (2020). Remote sensing for agricultural applications: A meta-review. Remote Sensing of Environment, 236, 111402.

---

## Author Comment (AC1)

**Response to comments by Dr. David Chaparro**

Dear Dr. David Chaparro,

We sincerely thank you for your quality review of our manuscript. Your constructive comments and suggestions are very useful to improve the quality of the paper. Please find below our point-to-point responses to your comments as well as the corresponding changes in the manuscript.

Sincerely,

Hong Zhao, Yijian Zeng, Jan G. Hofste, Ting Duan, Jun Wen, Zhongbo Su

Referee comments are written in normal font; author's responses are written in blue font; and proposed changes are highlighted in green font.

Review of "Modelling of Multi-Frequency Microwave Backscatter and Emission of Land Surface by a Community Land Active Passive Microwave Radiative Transfer Modelling Platform (CLAP), by Zhao et al.

**Reviewer: David Chaparro**

This paper presents a multifrequency active-passive transfer modelling, named CLAP, for simulating backscatter and emission signals of a vegetated (herbaceous) surface in the Maqu area in Tibet. Results are compared with ground-based radiometer and scatterometer measurements, and with AMSR2 data. The authors explore how simulations using different frequencies (L-, S-, C- and X- bands), vegetation configurations (cylinders vs. discs), seasons (winter vs. summer) and soil moisture sources (in situ vs modelled) track radiometer and scatterometer observations. With this, they aim at enhancing our understanding of vegetation, soil, and temperature impacts on daily variations in emission and backscatter.

The paper is well written and encompasses a wide range of characteristics influencing model outputs, thus being complete research. The work improves previous papers estimating microwave emissions by including the backscatter component in the outputs. Overall, it is research with good potential, but I have relevant major concerns that must be thoroughly addressed through the manuscript, especially regarding that: (i) the metrics used in the results should be extended,

Response: Thanks a lot. The metrics including the correlation coefficient R and unbiased root mean square ubRMSE have been added in Tables 3, 4, 5, 6 and 7 and Tables S2 and S3 in the supplementary materials. As the model parameter calibration is not our main concern in this manuscript, you may notice that the obtained statistics shown in Tables 3, 4, 5, 6 and 7 and Tables S2 and S3 are worsened than those shown in the previous studies (Dente et al., 2014; Zheng et al., 2021). While the similar poor statistic value (e.g., 3.91 dB) is also reported by Vermunt et al. (2021), which used the coupled IME-TVG with the support of ground-based L-band radar backscatter measured for corn in Florida. We expect to do the model parameter optimization using for instance a simple data assimilation framework, but this is currently beyond the scope of this manuscript. We added section 4.4 to discuss the 'Limitations in this study' in the revised manuscript.

(ii) it is needed an extended discussion based on literature previously mentioned by the authors in the introduction,

Response: Thank you. As described in Introduction, the previous studies, for instance, Dente et al. (2014) and Zheng et al. (2021) used the coupled IEM with TVG models (IEM-TVG) and focused on calibrating model parameters based on either in situ or satellite observations. Using calibrated parameters, the calculated statistics shown in their studies are good enough for forward signal modelling and SM retrieval.

However, our focus in this study is to go through CLAP, namely the coupled ATS-AIEM-TVG model with a more physically-based consideration than the coupled IEM-TVG. As described in Introduction, the dielectric roughness parameterized in the ATS model is related to wavelength information and the surface status modulated by hydrometeorological conditions (Equations 4,5,6 in (Zhao et al., 2021) ). Additionally, it is known that the employed AIEM in this study uses a more complete expression of the single scattering terms than IEM does to keep the acceptable energy conservation for calculating bistatic scattering and further emission.

Specifically, we focused on using CLAP—a uniform active and passive multi-frequency observation operator, to investigate all factors and possible physical processes that influence the dynamics of the observed multi-frequency microwave signals. In particular, we investigated the impacts of different soil moisture and temperature profiles, vegetation structure and dynamics of vegetation water and temperature on the signal simulation, as you saw in the manuscript, which was not investigated in the previous studies (Dente et al., 2014; Zheng et al., 2021).

Accordingly, we did revisions in the introduction:

Lines 120-129: "It models dynamic dielectric roughness related to surface status modulated by hydrometeorological conditions, additionally, the modelled dielectric roughness scales with wavelength with maintaining physical consistency (Zhao et al., 2021). The ATS model integrated with the AIEM was further coupled with the TVG model for simultaneously modelling L-band scattering and emission of the overall vegetation-soil medium, and the simulated signals were closer to the observations and exhibit more dynamics than those simulated by the model without considering the ATS model (Zhao et al., 2021). As such, the coupled ATS-AIEM-TVG model has a more physically-based consideration with the improved ability in modelling signal dynamics than the fundamental IEM-TVG model."

Lines 143-147: "Despite the above advancement, it is noted that the previous studies (Dente et al., 2014; Zheng et al., 2021) assumed the grass leaf to be parameterized with a disc shape and be calibrated with the constant VWC. The role vegetation (e.g., its different shapes and diurnal changes in water content and temperature) plays in mechanistic microwave scattering and emission processes reflected in these (in situ) microwave observations has not been explored yet."

Lines 166-169: "However, the variation in dielectric properties of both the soil and vegetation due to the temperature effect, and how the signal diurnal variation (affected by these factors) differs in terms of frequencies and surface conditions are not fully investigated in the previous studies (Dente et al., 2014; Zheng et al., 2021)".

(iii) the paper needs putting the results obtained in the context of the current and future missions and retrievals (e.g., which could be the impact of the reported RMSE and Biases in SM retrievals?). The third point is especially important, as the authors emphasize the applicability of the model stating that it can mimic observations and that it could be applied to global scales. None of these are fully justified if the reader cannot understand how CLAP could be applied: how can it be transferred to global scale? Which could be the impact of model errors if it was to be applied in moisture estimates?

Response: Thank you. We focused on understanding the impacts of vegetation water content and temperature and soil moisture and temperature profiles on forward simulations of dynamics in emission and backscatter. We presented in this manuscript what we achieved and pointed out the direction in which we should put efforts in the next step. We have not stepped into the retrieval (of SM or vegetation parameter). Based on the investigation results shown in this manuscript, the mismatch still exists between the dynamics of modelled signals and the observations, especially during the soil freeze-thaw processes. This may indicate that some fundamentals, for instance, the abrupt phase change of topsoil water during the winter period that leads to big differences in dielectric property values and the resulting signal dynamics as reported by Lv et al. (2022) are not expressed with explicit physical formulations in CLAP. In future work, we will consider this and improve the current ATS setup with THE two-layer model considering soil conditions both in the surface layer and penetration layer. Moreover, the differences of the normalized  $\tau$  at multi-frequency exist in this case and the physical reason needs to be investigated. As such, we still need to make clear what is observed at multi-frequency and how to use it in a consistent way. These are directions for future efforts but beyond scope of this study.

Regarding applying CLAP on a large scale up to a global scale, we have ideas and are working on it. However, we agree with you that saying CLAP applied on the global scale is too advanced in this paper, and this manuscript is long enough. We delete the phase involving the 'global' word in the relevant sentence to eliminate such meaning.

We added section 4.4 to discuss the 'Limitations in this study'. We went through the Results and Conclusions and did corresponding revisions.

**"Section 4.4**

As investigation results shown in section 3, by considering dynamics (i.e., diurnal cycles) of vegetation water content and temperature as well as soil moisture and temperature, the dynamics of the ground-based observed microwave signals are interpreted and modelled to some extent. However, there are limitations in this study that lead to mismatches between modelled signals and the observations as shown in some results with less good performance metric values. Because there are no continuous measurements of VWC in this study, a sinusoidal function is assumed to estimate daily VWC. The assumed phase shift ( $\phi$  in Equation (1)) influences the phase characteristic of the modelled signals. In situ vegetation temperature data is also not available, and the value of vegetation temperature are not equal, because vegetation cools off through evaporation and warms up through irradiance. The value of vegetation temperature should be in between that of air temperature and surface temperature, and it is noted that the phase shift exists between air temperature and surface temperature (see Fig. 2b and Fig. 7b). Thus, the vegetation temperature surrogate in this study also accounts for mismatches (in both magnitude and phase) between modelled signals and the observations especially at higher frequencies. Additionally, the investigation results show that the

vegetation orientation influences the variation of cross-polarization signals at high frequencies (see section 4.1), while the grass morphology during the growth period is not considered in this study.

Regarding the soil part, the investigation results shown in section 3.3 (e.g., Figure 12b (i.e.,  $\sigma_{VV}^0$  at Cband estimated based on in situ SM at 2.5 cm and SM at 1 mm simulated by STEMMUS process model) indicate the necessity of considering the topsoil moisture information in modelling dynamics of the observed signals, and the accuracy of topsoil moisture information influences the adequate simulation of signal dynamics. While the deviation exists between the process modelled soil moisture and in situ measurement (see Figures S1 and S2), accurate topsoil moisture information is still difficult to obtain (either by a process model or in situ measurement). Moreover, as discussed in section 4.2, the explicit physical formulations that can parameterize the abrupt change in dielectric properties due to the soil-water-ice phase change are not yet considered. Furthermore, surface roughness is another important factor influencing soil scattering and emission. The used values of the surface roughness parameter during the summer period in this study are calibrated values based on satellite observations. On the other hand, surface roughness during the winter period may exhibit slight changes due to the soil freeze-thaw processes, such as frozen soil water causing volume expansion and melted surface water smoothing the surface. These kinds of effects that influence the observed signal dynamics are also not considered in this study.

As the model parameter calibration is not our main concern in this paper, the obtained statistics (see Tables 3, 4, 5, 6 and 7 and Tables S2 and S3) are less satisfactory than those shown in the previous studies (Dente et al., 2014; Zheng et al., 2021). The similar poor performance of IEM-TVG simulation results (e.g., RMSE of 3.91 dB) is also reported by Vermunt et al. (2021), when compared with the ground-based L-band radar backscatter measured for corn in Florida. The model parameter optimization using for instance a simple data assimilation framework, is expected to help improve performance."

Lines 716-717: "Nevertheless, using microwave signals to probe leaf water potential can help understand plant water use regulation and the associated drought vulnerability at a regional scale."

Lines 829-833: "CLAP will be extended as a full-spectrum (from optical, thermal infrared, to microwave) observation operator by adding SCOPE modelling canopy radiative, photochemical, and physiological processes and STEMMUS modelling soil water, heat and vapor transfer processes. As such, CLAP has the potential to synergistically utilize available and future satellite resources for monitoring (e.g., vegetation and soil) variables of interest."

I detail my comments hereafter. Major comments

• The results presented are based on RMSE and Bias metrics between the estimates and the in situ, as well as on time-series plots. However, as the authors focus on the study of the daily variability, correlation metrics are needed. Also, optionally, scatters between in situ and estimates can be plotted next to the time-series (instead of in separate figures; note that the number of figures in the paper is very large and should not be increased).

Response: Thanks a lot. We've added R values in Tables 3, 4, 5, 6 and 7 and previous Tables S2 and S3 in the supplementary materials. As we mentioned in the previous reply, the limitations described in section 4.4 in this study lead to some poor statistic values between simulated signals and the collected

detailed ground-based observations, although in contrast, better statistical values (e.g., much higher R values) are obtained when compared to the satellite AMSR2 observations (see Tables 3, 4, 5 and 6). Our major focus in this manuscript is to investigate the dynamics of VWC and vegetation temperature and soil moisture and temperature on the modelled signals. We did not add scatter plots because we think they do not provide extra useful information in terms of current time series plots and statistic values. We anyhow attached the plotted scatter plots for your reference.

---

## Author Comment (AC2)

**Response to comments by Referee 2**

Dear Reviewer,

We sincerely thank you for your quality review of our manuscript. Your constructive comments and suggestions are very useful to improve the quality of the paper. We apologize for our imprecise expressions. We revised the Results and Conclusions to highlight the main contribution of this study based on what we have investiagted. We added the discussion on the limitations of this study to point out the directions for future efforts. Please find below our point-to-point responses to your comments, as well as the corresponding changes in the manuscript.

We hope that you can reconsider our revised manuscript.

Sincerely,

Hong Zhao, Yijian Zeng, Jan G. Hofste, Ting Duan, Jun Wen, Zhongbo Su

Referee comments are written in normal font; author's responses are written in blue font; and proposed changes are highlighted in green font.

The authors present a radiative transfer modelling platform for simulating backscatter and brightness temperature emission. They apply this model platform to simulate in-situ backscatter and emission data from a grassland site on the Tibetan plateau based on observed and simulated data for soil moisture and temperature. The authors investigate multiple different parameterizations target variables, and frequencies. A particular focus is put on explaining diurnal variability, where the authors try to disentangle the effects of diurnal variations of temperature and vegetation water content.

I believe that a better understanding of backscatter and brightness temperature and how they are influenced by diurnal variations of VWC and temperature can advance the scientific progress of the hydrological community. However, in the current state, I would advise the editor to reject the manuscript.

The manuscript has 3 major general issues, and also some other major issues with specific analyses performed, as listed below.

**Major general issues:**

- The article is too long, and there are too many figures (19 figures in the article, 27 figures in the supplement). The authors should put more effort into presenting their results in fewer figures using higher-level summaries of the obtained results.

Response: Thanks a lot for your suggestions. Supported by the detailed ground-based observations at the Maqu site, we tried our best to present in this manuscript our investigation results of CLAP—a unified multi-frequency scattering and emission observation operator, in simulating both backscatter and emission signals of grassland. The comprehensive investigations involve in different frequencies (L-, S-, C- and X-bands), polarization configurations (VV, HH, VH at active case and both V and H at passive case), vegetation configurations (cylinders vs discs structures; constant vs dynamic VWC and temperature) and soil states (in situ vs process modelled moisture and temperature).

As shown in Figures 2-5, the observed signals at each frequency reflect the surface status in their own unique way. For instance, the fluctuations of the observed signals at high frequencies (i.e., X- and C-bands) are more related to the vegetation part, while the dynamics of the observed signals at low frequencies (i.e., S- and L-bands) are more related to the soil part. The details of separate findings are described in the Results in the manuscript. We think that analyzing and modelling all these multi-frequency signals of the same scene helps us dive into the microwave challenge: to be able to describe and separate the contributions of the different components in the observed total signature of the vegetated lands. Moreover, by utilizing all these multi-frequency signals, the reader can see that we are trying to figure out a system that can model the dynamic signal due to the dynamic surface status (e.g., vegetation growing period, soil freeze-thaw period) under different hydrometeorological conditions (e.g., rainfall and no rainfall), although there are limitations in this study (please see new section 4.4 that we added).

By looking into figures that display time series observed and modelled signals, the reader can see interesting physical processes that influence signal dynamics and the ability of the model consideration to mimic signals. For instance, the soil freeze-thaw along the profile can be 'seen' by analyzing multi-frequency signals during the winter period. By investigating the impact of different soil states (in situ vs process-modelled moisture and temperature) on the modelled signals compared to the observations, the reader can see, for instance, the necessity of considering the topsoil moisture information in the modelling and considering possible physical formulations to parameterize the abrupt change in dielectric properties due to the soil-water-ice phase change. Although we have not focused on modelling signal variations due to rainfall events, based on displayed figures, the reader can see that small rainfall (e.g., 23/07/2018 in Figure 2) can be related to high jumps in the observed signals at X-band at VV and VH polarizations. Because of the less heavy rainfall amount, the other observed low frequencies signals do not reflect this. While heavy rainfall (e.g., 31/07/2018 in Figure 2) is 'seen' through high jumps in all frequency signals (under different polarizations though). Based on what we had and did, we'd like to convey to readers comprehensive investigation results and the directions for future efforts in this manuscript.

We wrote section 3.3, kept the most important figures for the winter case that mainly support our conclusions and put other figures in the supplementary materials for reference. In the end, the revised version has 13 figures in the text.

We deleted the previous Figure S1, as the shape of VWC is also plotted in Figure 2b. We kept the previous Figure S16 as a case to show that soil contribution $\sigma_s^0$ is dominant at X-band during the winter period in the Maqu case. The previous Figures 17-19 showing similar findings at C-, S- and L-bands were deleted. We kept other figures in the supplementary materials, as they provide extra details to support our demonstrations in the text and are also useful for experts in the field for reference. Although the amount of figures in the supplementary materials is large, as you see that some figures,

for instance at X-, C-, S- and L-band cases, can be labelled as subfigures to shorten the figure number. In order not to make one figure too heavy, we prefer to show the figure at each frequency one by one. We provide an overview of all figure information in the supplementary materials, and the reader can easily get the information and relocate the figure according to his/her demands.

- Throughout the manuscript, the authors state that CLAP can reproduce the observed signals. Based solely on the results shown, I am not convinced whether this is true. The authors should at least discuss in more detail why such large deviations as shown in the plots still qualify as "reproducing the observed signal".

Response: Many thanks for this point. The analysis shows potential for matching all active and passive channels by CLAP, but extra investigations are necessary to mitigate the limitations in future efforts. We added section 4.4 of 'Limitations in this study'. We also added metric values as references for good and poor performances described in the manuscript. We went through the Results and Conclusions and did corresponding revisions. The specific modifications in the text are listed below,

"Section 4.4

As investigation results shown in section 3, by considering dynamics (i.e., diurnal cycles) of vegetation water content and temperature as well as soil moisture and temperature, the dynamics of the ground-based observed microwave signals are interpreted and modelled to some extent. However, there are limitations in this study that lead to mismatches between modelled signals and the observations as shown in some results with less good performance metric values. Because there are no continuous measurements of VWC in this study, a sinusoidal function is assumed to estimate daily VWC. The assumed phase shift ($\phi$ in Equation (1)) influences the phase characteristic of the modelled signals. In situ vegetation temperature data is also not available, and the value of vegetation temperature is assumed to be the same as that of air temperature. While it is known that vegetation temperature and air temperature are not equal, because vegetation cools off through evaporation and warms up through irradiance. The value of vegetation temperature should be in between that of air temperature and surface temperature, and it is noted that the phase shift exists between air temperature and surface temperature (see Fig. 2b and Fig. 7b). Thus, the vegetation temperature surrogate in this study also accounts for mismatches (in both magnitude and phase) between modelled signals and the observations especially at higher frequencies. Additionally, the investigation results show that the vegetation orientation influences the variation of cross-polarization signals at high frequencies (see section 4.1), while the grass morphology during the growth period is not considered in this study.

Regarding the soil part, the investigation results shown in section 3.3 (e.g., Figure 12b (i.e., $\sigma_{VV}^0$ at C-band estimated based on in situ SM at 2.5 cm and SM at 1 mm simulated by STEMMUS process model) indicate the necessity of considering the topsoil moisture information in modelling dynamics of the observed signals, and the accuracy of topsoil moisture information influences the adequate simulation of signal dynamics. While the deviation exists between the process modelled soil moisture and in situ measurement (see Figures S1 and S2), accurate topsoil moisture information is still difficult to obtain (either by a process model or in situ measurement). Moreover, as discussed in section 4.2, the explicit physical formulations that can parameterize the abrupt change in dielectric properties due to the soil-water-ice phase change are not yet considered. Furthermore, surface roughness is another important factor influencing soil scattering and emission. The used values of the surface roughness parameter during the summer period in this study are calibrated values based on satellite observations.

On the other hand, surface roughness during the winter period may exhibit slight changes due to the soil freeze-thaw processes, such as frozen soil water causing volume expansion and melted surface water smoothing the surface. These kinds of effects that influence the observed signal dynamics are also not considered in this study.

As the model parameter calibration is not our main concern in this paper, the obtained statistics (see Tables 3, 4, 5, 6 and 7 and Tables S2 and S3) are less satisfactory than those shown in the previous studies (Dente et al., 2014; Zheng et al., 2021). The similar poor performance of IEM-TVG simulation results (e.g., RMSE of 3.91 dB) is also reported by Vermunt et al. (2021), when compared with the ground-based L-band radar backscatter measured for corn in Florida. The model parameter optimization using for instance a simple data assimilation framework, is expected to help improve performance."

The added metric values as references for good and poor performances described in the text.

Lines 428-432: "Figure 6 shows that $T_B^H$ simulated with the disc parameterization is closer to ELBARA-III observed $T_B^H$ at L-band (RMSEs of 12.7 vs 30.4 K in Table 4) than those with the cylinder parameterization, despite slightly higher R values obtained by cylinder parameterization (R of 0.27 vs 0.16). While the disc parameterization performs similarly as the cylinder parameterization does in the good simulation of $T_B^V$ (Fig. 6, and R over 0.8 and RMSEs of 6.5 K and 5.6 K in Table 4)".

Lines 487-493: "Figure S19 shows that using STEMMUS simulated soil moisture and temperature as the input in CLAP does not outperform using the in situ measured soil moisture and temperature in reproducing the observed $\sigma_{pq}^0$ at X-band, and both lead to the heavy overestimation of $\sigma_{HH}^0$ at X-band with poor performance metric values (e.g., Biases of 10.9 dB and RMSE of 11.0 in Table 5 for in situ case). However, using STEMMUS simulated soil moisture and temperature outperforms in reproducing the observed $\sigma_{pq}^0$ at C-band (Fig. 8) with good performance metric values of Bias of 1.6 dB and RMSE of 1.9 dB (Table 6)."

Lines 531-534: "Figure 9 also shows that the model can capture the observed diurnal variations (with R of 0.65 and 0.73 for H and V polarization respectively, Table 7), but the large systematic underpredictions of $T_B^p$ are observed in comparison to the ELBARA-III observations (RMSEs of 26.3 K under V polarization and 48.9 K under H polarization in Table 7)."

Lines 636-639: "Figure 12 shows that the Case1_surface_moisture_shallow_boundary can capture the observed diurnal variation of $T_B^p$ signals but present a systematic overprediction (RMSEs of 15.0 K at V polarization and 16.5 K at H polarization in Table S4) of $T_B^p$, which is opposite to the heavy underprediction by Case0_surface_moisture_deep_boundary (RMSEs over 26 K in Table 7)."

Lines 648-652: "In contrast, the Case2_penetration_depth_moisture_shallow_boundary captures the magnitude of the observed $T_B^H$ (RMSE of 5 K in Table S4) and $\sigma_{VV}^0$ (RMSE of 0.9 dB in Table S5), the Case3_penetration_depth_moisture_boundary simulates the magnitude of the observed $T_B^V$ (RMSE of 5 K in Table S4) and $\sigma_{HH}^0$ (RMSE of 3.3 dB in Table S5), although the diurnal changes simulated by both cases are flat."

In Conclusions: "In comparison to the in situ and available satellite AMSR2 X- and C-bands microwave emission observations during the summer period, CLAP using the cylinder

parameterization of vegetation representation can mimic multi-frequency $\sigma^0_{pq}$ better than the disc parameterization does (e.g., mean RMSEs of 2.1 vs 3.7 dB, with L-band $\sigma^0_{VH}$ data excluded), and it performs similarly well as the disc parameterization does in the simulation of L-band $T^V_B$ (e.g., with RMSEs around 5.6 K and R over 0.8), although the latter case performs better in the simulation of $T^H_B$ (e.g., with RMSEs of 12.7 K)……

The comparison results show that CLAP using the cylinder parameterization and either the in situ measurements or the process model outputs can mimic the observed C-band co-polarization $\sigma^0$ of grassland especially at VV polarization (e.g., with RMSE of 1.9 dB) and its diurnal variations during the winter period. However, the current platform cannot reproduce $\sigma^0$ and $T^p_B$ dynamics at other bands during this period."

- As has also been pointed out by the other reviewer, the used metrics are not suitable to evaluate the diurnal or day-to-day variations. Metrics like correlation, unbiased RMSE, or even a detailed analysis of the mean diurnal cycles regarding magnitude or phase shift should be included.

Response: Thanks a lot. We added statistics of correlation and unbiased RMSE in Tables 3, 4, 5, 6 and 7. Regarding the detailed analysis of the mean diurnal cycles, we do not have continuous VWC measurements to support it in this manuscript.

**Major specific issues:**

- The authors compare the use of in-situ SM from 2.5cm depth with modelled SM from 1mm depth, and from this draw conclusions about the sensing depths of the different bands. The effect mainly shows as a bias component in the simulated backscatter, while the differences in the diurnal patterns are visible, but small. However, as shown in Fig. S2, there is a bias between modelled and in-situ soil moisture even at similar depths. The different depths in the model all show the same "base level" and differ only in the magnitude of their diurnal variations. The difference in absolute value between using in-situ 2.5cm depth and modelled 1mm SM (e.g. in Fig. 11) therefore seems to be more related to this bias, than to difference in depth. Additionally, there is also a difference in the magnitude of diurnal variations between model and observations. The variations in the model at similar depths are much more pronounced, which furthermore makes the model simulations and the in-situ observations hard to compare. I would therefore be careful about drawing any conclusions about sensing depths from these comparisons.

Response: Thanks a lot. We think that these two mentioned issues (i.e., the differences related to bias and depth) cannot be separated. We were trying to link physical quantities to explain the observed signal dynamics. The sensing depth in the soil does exist and varies in terms of frequency and surface conditions. Since the X- and C-band EM waves exhibit shorter wavelengths than L-band EM waves, in theory, their sensing depths should be lower than that at L-band. In situ measured topsoil moisture is not available, but the process-based model can provide simulated values of the topsoil. In this manuscript, we used what we have to discuss this issue. We added the caution you suggested in section 4.4 of 'Limitations in this study'.

- Fig. S12: I have some trouble understanding why Case 3 and Case 4 show such strong differences in mean and do not even cross. Since Case 4 uses the same air temperature, but imposes a diurnal pattern

on VWC, I would guess that they have to be the same at least twice a day. All other plots also show that Case 3 and Case 4 are closely together and only show diurnal differences.

Response: Thank you for spotting this. We apologize for our mistake. Case 4 is under the incidence angle of 50°. We put the right results under the incidence angle of 40° at L-band in revised Figure S12. Figure 5 in the manuscript was also revised.

[Figure]

*Figure S11 $\sigma_{pq}^0$ at L-band simulated by the CLAP (ATS_AIEM_TVG model) with the disc parameterization using three different cases of vegetation temperature and VWC, compared to the ground-based observations during the summer period. Case3 refers to dynamic vegetation temperature (i.e., in-situ measured air temperature at 2 m) but constant VWC (0.6 kg/kg). Case4 refers to the dynamic vegetation temperature used in Case3 and the estimated dynamic VWC (Figure 2b in the text).*

[Figure]

*Figure 5 $\sigma_{pq}^0$ at L-band simulated by CLAP with the disc and cylinder parameterizations compared to the ground-based observations during the summer period.*

- In lines 508-510 the authors claim that observed winter-period VV diurnal variations of backscatter can be reproduced with CLAP. With some imagination I can see this in C-band, but not in the other bands. Please rephrase or show other plots or metrics supporting this statement.

Response: Thank you. We have rewritten section 3.3. The revised sentences are in

Lines 509-515: "Both simulated co-polarization $\sigma_{pq}^0$ at S- and L-bands do not exhibit pronounced diurnal variations as the observations do (Figs. A5 and A6). In short, the observed co-polar $\sigma_{pq}^0$ at C-band and its diurnal variations especially at VV polarization during the winter period can be reproduced by CLAP using the process model simulated soil moisture and temperature as the input (i.e., the STEMMUS model in this case). Further investigations need to be done for improving CLAP simulations of the co-polar $\sigma_{pq}^0$ at other bands."

**Minor comments:**

- Setting the phase shift of the VWC curve to $\pi/2$ means that the minimum is reached at noon, and the maximum at midnight, but according to the description in section 2.2, VWC is replenished until early morning. This implies that the maximum should be at early morning. I don't think this is a large problem per se, because currently inferences are only made about the impact of dynamic VWC on modelled backscatter. But it might be something to consider in case a more detailed analysis of the mean diurnal cycles reveals a phase shift.

Response: Thanks a lot for your comments. We agree that the explicit timing information of VWC would contribute to explaining the phase difference between simulated signals and the observations. An example is shown in Figure 9 from Vermunt et al. (2022) based on their in situ measurements of a

maize field. In our case, we do not have continuous VWC measurements to support the detailed analysis of the mean diurnal cycles in this manuscript.

- Change "the CLAP" to "CLAP" throughout the manuscript. I also encourage the authors to google for "the clap".

Response: Thanks. Done.

- If feasible, it would be helpful to have (i) an overview table of all parameters going into the model (maybe in the appendix) and (ii) a short summary of the main model equations of the used models (TVG, AIEM, ATS), also in the appendix.

Response: Thanks a lot. The main model equations can be found in the cited manuscript. In order not to lengthen the manuscript, we added Table S2 to provide an overview of input parameters in CLAP and Table S1 with the main model equations used in the models in the supplementary materials.

Table S1 An overview of the main equations used in CLAP.

| Model | Main equations |
|-------|----------------|
| TVG | $\sigma = 4\pi\left|f_{pq}\right|^2$, where $\sigma$ (m²) denotes the polarized scattering cross-section $\sigma$ for a single scatter. $f_{pq}$ denotes the scattering function and can be calculated by the aforementioned electromagnetic approximation (see section 2.4 in the text). |
| | $\sigma_{eq} = \frac{4\pi}{k} Im\langle f_{qq}^F\rangle$, where $\sigma_{eq}$ denotes the extinction cross section of the scatter, and $f_{qq}^F$ is the average scattering amplitude calculated in the forward direction ($\theta_s = \theta$ and $\phi_s - \phi = 0$), k denotes the wave number. |
| | $S_{pqm} = n\Delta z \frac{\Delta\theta\sin(\theta_i)}{4\pi\cos(\theta_s)} a_m \mathcal{F}_m\{\sigma_{pq}(\phi_s - \phi)\}$, where $\boldsymbol{S}$ represents the vegetation scattering matrix. $n$ denotes the scatter density (m⁻³) and $\Delta z$ denotes the sublayer thickness (m). In the modelling, the number of scatters per unit area (m⁻²) is used to represent $n\Delta z$. The Fourier transform method ($a_m, \mathcal{F}_m$) is used to express the azimuthal dependence of scattering. $m$ denotes the number of series terms. $\sigma_{pq}$ denotes the average polarized bistatic scattering cross-sections in the half-space. |
| | $S_{gpqm} = \frac{\Delta\theta\sin(\theta_i)d\phi}{4\pi\cos(\theta_s)} a_m \mathcal{F}_m\{\sigma_{gpq}^0(\phi_s - \phi)\}$, where $\boldsymbol{S_g}$ represents soil scattering matrix. $\sigma_g^0$ denotes the bistatic scattering coefficient and is calculated by AIEM. |
| | $\sigma_{pq}^0 = \frac{4\pi}{\Delta\theta}\frac{\cos(\theta_i)}{\sin(\theta_i)} S_{pq}'(\pi)$, where $\boldsymbol{S'}$ denotes the combined vegetation-soil matrix through the Fourier inverse transform. |
| | $\varepsilon = 1 - \sum_{i=1}^{2N_\theta} \frac{\cos(\theta_s)\sin(\theta_s)}{\cos(\theta_i)\sin(\theta_i)} S_0'$, where $\boldsymbol{S_0'}$ denotes the zeroth order components of the combined matrices. $\boldsymbol{N_\theta}$ refers to the angular resolution in off-nadir angle $\boldsymbol{\theta}$ direction (see Table 2 in the text). |

| | |
|---|---|
| AIEM | $$\sigma_{gpq}^0 = \sigma_{pq}^k + \sigma_{pq}^{kc} + \sigma_{pq}^c$$ $$= \frac{k^2}{2}\exp\left[-s^2\left(k_z^2 + k_{sz}^2\right)\right]\cdot\sum_{n=1}^{\infty}\frac{s^{2n}}{n!}\left|I_{pq}^n\right|^2 W^{(n)}\left(k_{sx} - k_x, k_{sy} - k_y\right)$$ *With* $$I_{pq}^n = (k_{sz} + k_z)^n f_{qp} e^{-\sigma^2 k_z k_{sz}} +$$ $$\frac{1}{4}\begin{cases}(k_{sz} - q_1)^n F_{pq1}^{(+)} e^{-s^2(q_1^2 - q_1 k_{sz} + q_1 k_z)} + \\ (k_{sz} - q_2)^n F_{pq2}^{(+)} e^{-s^2(q_2^2 - q_2 k_{sz} + q_2 k_z)} + \\ (k_{sz} + q_1)^n F_{pq1}^{(-)} e^{-s^2(q_1^2 + q_1 k_{sz} - q_1 k_z)} + \\ (k_{sz} + q_2)^n F_{pq2}^{(-)} e^{-s^2(q_2^2 + q_2 k_{sz} - q_2 k_z)}\end{cases}\bigg| u, v = -k_x, -k_y +$$ $$\frac{1}{4}\begin{cases}(k_z + q_1)^n F_{pq1}^{(+)} e^{-s^2(q_1^2 - q_1 k_{sz} + q_1 k_z)} + \\ (k_z - q_2)^n F_{pq2}^{(+)} e^{-s^2(q_2^2 - q_2 k_{sz} + q_2 k_z)} + \\ (k_{sz} - q_1)^n F_{pq1}^{(-)} e^{-s^2(q_1^2 + q_1 k_{sz} - q_1 k_z)} + \\ (k_z + q_2)^n F_{pq2}^{(-)} e^{-s^2(q_2^2 + q_2 k_{sz} - q_2 k_z)}\end{cases}\bigg| u, v = -k_{sx}, -k_{sy}$$ *where $\sigma_{gpq}^0$ denotes the bistatic scattering coefficient. $\sigma_{pq}^k$, $\sigma_{pq}^{kc}$ and $\sigma_{pq}^c$ refer to the Kirchhoff, cross and complementary terms respectively. $s$ is standard deviation of height. $W^{(n)}$ is the Fourier transform of the nth power of the normalized surface correlation function. $k$ denotes the incidence wavenumber. $\overrightarrow{k_\iota} = \{k_x, k_y, k_z\}$ and $\overrightarrow{k_s} = \{k_{sx}, k_{sy}, k_{sz}\}$ denote wave vectors in incident and scattering directions, respectively. $f_{qp}$ denotes the Krichhoff field coefficient. When dealing with complementary field $F_{qp}$ coefficients, the reradiated fields propagate through medium 1, denoted by $F_{pq1}^{(+)}$ and $F_{pq1}^{(-)}$, and through medium 2, denoted by $F_{pq2}^{(+)}$ and $F_{pq2}^{(-)}$. $q_1$ and $q_2$ represent the surface heights at different locations on the random surface. Detailed explanations can be found in Chen et al. (2003).* |
| ATS | $h = h_{SS} + h_{SV}$, *where h is a dielectric roughness thickness characterizing the depth of interfaces, not only resulting from topsoil structures ($h_{SS}$) affected by both irregularities (i.e., geometric roughness) of the soil surface and inhomogeneous distribution of moisture, but also due to inhomogeneity within soil volume ($h_{SV}$) that is related to soil porosity and moisture.* |
| | $$h_{SS} = \begin{cases}2\cdot s\cdot(-\ln(SM))\cdot\cos^{Np}(\theta) & SM < FC \\ 2\cdot s & SM \geq FC\end{cases}$$ $$h_{SV} = \frac{-\ln(\text{soil porosity})}{\alpha}, \alpha = \frac{2\pi}{\lambda_0} * \frac{\varepsilon_{soil}''}{\sqrt{\varepsilon_{soil}'}}$$ $$\varepsilon(z^*) = \varepsilon_{air} + \frac{1}{1 + \exp\left(-\frac{z^* - h_{SV}}{k_{AS}}\right)}(\varepsilon_{soil} - \varepsilon_{air}), k_{AS} = \exp(-\alpha z^*)\cdot s$$ *where $\varepsilon(z^*)$ is the constructed dielectric profile from air to bulk soil medium. $s$ is the standard deviation of heights. SM is volumetric soil moisture and $\theta$ is the incidence angle. Np is a polarization modulation parameter, and Np is set at 0 for H polarization and -1 for V polarization. FC refers to field capacity. $\varepsilon_{soil} = \varepsilon_{soil}' + i\varepsilon_{soil}''$, $\varepsilon_{soil}'$ is real part, and $\varepsilon_{soil}''$ imaginary part of the bulk soil dielectric constant.* |

Table S2 An overview of input parameters in CLAP.

| | Parameter name | Values |
|---|---|---|
| TVG: Vegetation part | Vegetation morphology parameters | Table 2 in the text |
| | Number of vegetation scatters | |
| | Vegetation water content (VWC) | Estimated VWC in summer through Eq.(1) in the text; Assumed 0.04 (kg/kg) of VWC in winter. |
| | Vegetation temperature | In situ air temperature |
| AIEM+ATS: Soil part | Volumetric soil moisture and soil temperature | In situ measurements at 2.5, 5, 10, 20, 35 and 60 cm; STEMMUS simulated profiles at 1, 2, 5, 10, 15, 20, 30, 40, 50 and 60cm |
| | Soil texture | In situ measurements |
| | Standard deviation of height (cm) | 0.9 cm in summer and 0.4 cm in winter |
| | Correlation length (cm) | 9 cm in summer and 12 cm in winter |
| | Autocorrelation function | Exponential |
| Sensor configuration | Incidence angle (°) | Table 1 in the text |
| | Frequency | |
| | Polarization | |

- l. 55: "instruments ... measure"

Response: Thanks. Done.

- l. 63: the presence of vegetation

Response: Thanks. Done.

- l. 64: needed -> necessary

Response: Thanks. Done.

- l. 77: noticeable -> noteworthy?

Response: Done.

- l. 78: effect of vegetation on *the* microwave signal

Response: Done.

- l. 102: "the ground truth observations" is a bit vague, do you mean in-situ

backscatter/emission as you used in this paper?

Response: Yes. We have revised the sentence in

Line 106: "in comparison to in situ measured backscatter and emission".

- l. 132: how the vegetation plays the role in -> the role vegetation plays in

Response: Thanks. Done.

- l. 133: is not explored yet -> has not been explored yet

Response: Done.

- l. 140: "imposes great impacts on variations of sampling depth": unclear what you want to say here

Response: We are sorry for the ambiguity. We mean the variations of penetration/emission depth in the soil. We revised the sentence in

Line 160: "imposes great impacts on variations of penetration/emission depth in the soil, especially for high-frequency signals".

- l. 147: maybe start a new paragraph here with "It is well established..."

Response: Thanks. Done.

- l. 153: temeprature -> temperature

Response: Done.

- l. 158: forward simultaneous simulations -> simultaneous forward simulations

Response: Done.

- l. 190: they exhibit *a* difference

Response: Done.

- l. 201: for use -> used

Response: Done.

- l. 217: is valued at -> is set to (also l. 219)

Response: Done.

- l. 229: "comparable to the in situ measurements": See my comment above, there are biases and it is not clear how these influence your results. Did you calibrate the soil parameters for the site?

Response: Thanks. The 5TM probe sensor calibration was done for soil moisture measurements by Dente et al. (2012). To make it clear, we added a brief sentence in

Line 194: "Specific calibrations were conducted for the profile soil textures (Dente et al., 2012)".

- l. 240-244: This sentence is very long, it would help readability to split it into multiple sentences.

Response: Thank you. The sentence was revised in

Lines 254-260: "After defining the shape of discrete scatters, in respect of selected geometry and frequency (Fig. 1), the electromagnetic approximations (i.e., Rayleigh-Gans (Eom & Fung, 1984; Schiffer & Thielheim, 1979), physical optics (LeVine et al., 1983) and infinite length (Karam & Fung, 1988; Wait & Maxwell, 1988)) are adopted correspondingly to calculate vegetation bistatic scattering and extinction (absorption plus scattering) cross-sections. Therein the vegetation dielectric constant is calculated from either the Matzler (1994) model (gravimetric VWC not less than 0.5 (kg/kg)) or Ulaby (1987) model (dry vegetation)."

- l. 248: for obtaining *the* soil scattering matrix

Response: Thanks. Done.

- l. 260: remove "below (Eq. (3))"

Response: Done.

- l. 286-287: remove "please also refer to"

Response: Done.

- l. 312: The role of the low air pressure in the approximation is not clear to me, can you make this more detailed?

Response: Thank you. Values of vegetation temperature should lie in between those of air temperature and those of surface temperature. In Tibetan Plateau, there is strong solar radiation, and the plant should get an intense warm-up. While due to low pressure, which correlates with low air temperature, plant temperature cannot get as high as land surface temperature. As such, air temperature is closer to vegetation temperature, and values of air temperature can be assigned to values of vegetation temperature. We revised the sentence in

Line 330: "the value of air temperature measured at 2 m above the surface (Fig. 2b ) is assigned to the value of vegetation temperature, which is acceptable because of the low air pressure on the Tibetan Plateau, where vegetation regulates temperature close to air temperature by transpiration".

- l. 330: What is the reason for the low value? Does this indicate a problem with the model or the model parameters in this setup?

Response: Thanks a lot. The low value is mainly due to volume scattering effects, which are present in the soil and exaggerated by the presence of ice and snow but are not considered by the surface scattering model of AIEM. We added this sentence in lines 347-349.

- l. 333: simulated at X-band

Response: Thanks. Done.

- l. 338-341: You show that both vary at similar frequencies, but is there a correlation between wind and X-band signal?

Response: The correlation coefficient is used to measure the strength of the linear relationship between two variables. However general sense is that the wind influences vegetation orientation and its angle distribution and further vegetation scattering and emission in a nonlinear way, which is hard to quantify as far as we know. Therefore, we did the Fourier transform analysis, which is in practice used in modelling nonlinear systems. Based on results in frequency domains, we tried to explore in which frequencies which factors may contribute to the variation of observed signals. As such, we approach to understand the possible reason that does not lie in the established assumptions in the model but may account for the mismatched variations between the observations and model simulation results (as shown in Figure 2).

Nevertheless, we calculated the correlation coefficient (R) between the observed $\sigma^0$ at X- and L-bands and wind speed. To assess the degree of correlation, we calculated R between the observed $\sigma^0$ at X-band and soil moisture at 2.5 cm (SM_2.5cm) as a reference, as surface soil moisture is known mainly contributing to soil scattering. The results show that with SM_2.5cm, $\sigma^0_{pq}$ at X-band exhibits R of 0.3 at VV polarization, R of 0.12 at HH polarization and 0.29 at VH polarization. Comparatively, with wind speed, $\sigma^0_{pq}$ at X-band exhibits R of 0.09 for VH polarization, R of 0.03 for VV polarization and R of 0.18 for HH polarization. In contrast, with wind speed, $\sigma^0_{pq}$ at L-band exhibits R of 0.05 for VH polarization, R of -0.0.2 for VV polarization and R of 0.05 for HH polarization. These R results indicate that the single factor (e.g., soil moisture and wind speed) exhibits a weak linear relationship with the observed $\sigma^0$ at X-band (if we applied the rule of thumb that a R of 0.35 represents a "weak" association). The complex physical process is there and we focus on combining the observed signals and physically based model platform to understand it. Considering our main focus in this study and the manuscript length, we will not add the above descriptions in the text.

- l. 398: despite -> except for?

Response: Thanks. Done.

- l. 458: while those simulated at L-band to not

Response: Done.

- l. 458: Is Figure 4 a wrong reference? Which figure should this refer to?

Response: Thanks a lot for spotting this. We are sorry for this mistake. Here it refers to Figure 7 and Figures A1-A3 in the shortened manuscript (Figures 7-10 in the old version).

- l. 462: the volume scattering effect *is* present

Response: Done.

- l. 480: remove "While"

Response: Done.

- l. 495: as mentioned above, I cannot see that the diurnal variations agree with the observations from the plots

Response: Thanks a lot for spotting this. We are sorry for this mistake. We revise the sentence into

Lines 509-515 "Both simulated co-polarization $\sigma_{pq}^0$ at S- and L-bands do not exhibit pronounced diurnal variations as the observations do (Figs. A5 and A6). "

- l. 517-518: The cylinder parameterization outperforms the disc parameterization in simulating ...

Response: Thanks a lot. Done.

- l. 523: deep -> low

Response: Done.

- l. 535: despite -> with?

Response: Done.

- l. 543: Are the values for L-band exactly zero or just very small?

Response: In this case, CLAP with the cylinder parameterization estimates $\omega$ at L-band with exact zero values.

- section 4.2: a more verbose naming scheme for the different cases might help understanding your argument here

Response: Thanks a lot. This part discusses different considerations of parameterizing the contribution from the surface and beneath soil at the emission depth. The parameters considered in the parameterization involve soil moisture, surface roughness s and wavelength information. Regarding the input of soil moisture, the related question is soil moisture at which layer should be used for constructing the dielectric profile from air to bulk soil medium. Accordingly, which dielectric layers determine the whole reflectivity of the composite air-soil medium and represent the penetration depth of soil moisture, a similar concept as in Wilheit (1978). Based on what we had, for instance, soil moisture measured at 2.5 cm and other depths and the detailed observed TB signals, we did investigations. The results displayed in Figure 18 demonstrate the necessity of considering

contributions from the surface and beneath soil at the emission depth, however, the dynamics in the observed TB still cannot be reproduced by the current set-up. There are still issues, for instance, the explicit physical formulations on abrupt soil water phase change and the topsoil moisture are not considered yet and need further investigations. With explicit consideration of the surface layer, the current ATS model can be adjusted into a two-layer model for practical simplicity, but these are beyond this study. We have put these descriptions in section 4.2 to make it clear.

We renamed cases as "Case0_surface_moisture_deep_boundary, Case1_surface_moisture_shallow_boundary, Case2_penetration_depth_moisture_shallow_boundary and Case3_penetration_depth_moisture_boundary."

- l. 664: To me it seems not really useful to use CLAP for detecting rainfall events, if they could also be detected directly from the data going into CLAP (e.g. the soil moisture data)

Response: Thanks a lot. It is indeed that the measured SM at 2.5 cm in our case can be used to directly detect rainfall events. However, topsoil moisture is not always available. Supposing that we only have SM measured at 5 cm, the variation of SM on 23/07/2018 shown in Figure 2 is not sufficient to assure the precipitation event. But the compared large difference between CLAP simulated results and both passive and active observations at X-band can indicate rainfall occurring.

- l. 689: While as -> Whereas
Response: Thanks. Done.

- l. 691: varied -> different
Response: Done.

- l. 740: observed signal -> modelled signal, since your dynamic VWC only influences the modelled signal

Response: Thanks for spotting this. Done.

Reference

Chen, K.-S., Wu, T.-D., Tsang, L., Li, Q., Shi, J., & Fung, A. K. (2003). Emission of rough surfaces calculated by the integral equation method with comparison to three-dimensional moment method simulations. *IEEE Transactions on Geoscience and Remote Sensing, 41*(1), 90-101. doi:10.1109/TGRS.2002.807587

Dente, L., Ferrazzoli, P., Su, Z., van der Velde, R., & Guerriero, L. (2014). Combined use of active and passive microwave satellite data to constrain a discrete scattering model. *Remote Sens. Environ., 155*, 222-238. doi:10.1016/j.rse.2014.08.031

Dente, L., Vekerdy, Z., Wen, J., & Su, Z. (2012). Maqu network for validation of satellite-derived soil moisture products. *Int. J. Appl. Earth Obs. Geoinf., 17*, 55-65. doi:10.1016/j.jag.2011.11.004

Eom, H. J., & Fung, A. F. (1984). A scatter model for vegetation up to Ku-band. *Remote Sens. Environ., 15*(3), 185-200. doi:10.1016/0034-4257(84)90030-0

Karam, M. A., & Fung, A. K. (1988). Electromagnetic scattering from a layer of finite length, randomly oriented, dielectric, circular cylinders over a rough interface with application to

vegetation. *International Journal of Remote Sensing, 9*(6), 1109-1134. doi:10.1080/01431168808954918

LeVine, D., Meneghini, R., Lang, R., & Seker, S. (1983). Scattering from arbitrarily oriented dielectric disks in the physical optics regime. *JOSA, 73*(10), 1255-1262.

Matzler, C. (1994). Microwave (1-100 GHz) dielectric model of leaves. *IEEE Trans. Geosci. Remote Sens., 32*(4), 947-949. doi:10.1109/36.298024

Schiffer, R., & Thielheim, K. O. (1979). Light scattering by dielectric needles and disks. *Journal of Applied Physics, 50*(4), 2476-2483. doi:10.1063/1.326257

Ulaby, F. a. E.-R., M. . (1987). Microwave dielectric spectrum of vegetation - part II: Dual dispersion model. *IEEE Trans. Geosci. Remote Sens., 25*, 550-556. doi:10.1109/TGRS.1987.289833

Vermunt, P. C., Khabbazan, S., Steele-Dunne, S. C., Judge, J., Monsivais-Huertero, A., Guerriero, L., & Liu, P.-W. (2021). Response of Subdaily L-Band Backscatter to Internal and Surface Canopy Water Dynamics. *IEEE Trans. Geosci. Remote Sens., 59*(9), 7322-7337. doi:10.1109/TGRS.2020.3035881

Vermunt, P. C., Steele-Dunne, S. C., Khabbazan, S., Judge, J., & van de Giesen, N. C. (2022). Extrapolating continuous vegetation water content to understand sub-daily backscatter variations. *Hydrology and Earth System Sciences, 26*(5), 1223-1241. doi:https://doi.org/

Wait, J. R., & Maxwell, R. (1988) Electromagnetic radiation from cylindrical structures: reprint edition. In*: Vol. 27. IEE Electromagnetic waves series*. United Kingdom: Institution of Electrical Engineers, Stevenage.

Wilheit, T. T. (1978). Radiative transfer in a plane stratified dielectric. *IEEE Trans. Geosci. Electron., 16*(2), 138-143. doi:10.1109/TGE.1978.294577

Zheng, D., Li, X., Wen, J., Hofste, J. G., van der Velde, R., Wang, X., . . . Su, Z. (2021). Active and Passive Microwave Signatures of Diurnal Soil Freeze-Thaw Transitions on the Tibetan Plateau. *IEEE Trans. Geosci. Remote Sens.*, 1-14. doi:10.1109/TGRS.2021.3092411